# Deletion of EP3 prostaglandin receptor in murine macrophages aggravates diet-induced obesity by suppressing SPARC

Wenlong Shang [1,5], Yinxiu Li [1,5], Lu Wang [2], Jiao Liu [1], Huiwen Ren [1], Qian Liu [1], Shumin Guo [1], Yuhong Wang [1], Yubo Ma [1], Tianyi You [2], Yujun Shen [1], Yu Zhou [3], Danyang Tian [1,4 ✉] & Ying Yu [1 ✉]

## Abstract

**Macrophages are primary immune cells involved in obesity-triggered chronic low-grade inflammation in adipose tissues. Prostaglandin E2 (PGE$_2$), mainly generated from macrophages, can regulate adipose tissue remodeling, yet the underlying mechanisms are not fully understood. Here, we observed that PGE$_2$ receptor subtype 3 (EP3) was remarkably downregulated in adipose tissue macrophages from high-fat diet (HFD)-fed mice and patients with obesity. Notably, macrophage-specific deletion of EP3 exacerbated HFD-induced fat expansion, whereas EP3α isoform overexpression in macrophages alleviated obesity phenotypes. Further, EP3 deficiency suppressed secretion of anti-adipogenic matricellular protein SPARC from macrophages. SPARC deletion in macrophages abrogated the protection of EP3-overexpression against diet-induced obesity. Mechanistically, EP3 activation promoted SPARC expression by suppressing DNA methylation in macrophages through a PKA-Sp1-Dnmt1/3a signaling cascade. Finally, EP3 agonist treatment ameliorated HFD-induced obesity in mice. Thus, EP3 inhibits adipogenesis through promoting release of SPARC from macrophages, suggesting a novel therapeutic target for diet-induced obesity.**

**Keywords** Obesity; Macrophage; E-prostanoid 3 Receptor; SPARC
**Subject Categories** Cell Adhesion, Polarity & Cytoskeleton; Immunology; Metabolism

## Introduction

Obesity has emerged as an increasing global public health and social issue and is widely linked to metabolic and cardiovascular disorders, including type 2 diabetes, fatty liver disease, hypertension, heart failure, atherosclerosis (Elmaleh-Sachs et al, 2023). Pathological adipose tissue remodeling including adipocyte hypertrophy and apoptosis, immune cells infiltration, vascularization, and extracellular matrix deposition is a hallmark feature in the development of obesity (Marcelin et al, 2019). Obesity results in chronic systemic inflammation of adipose tissues and eventually leads to insulin resistance and type 2 diabetes (Rohm et al, 2022). Infiltrating macrophages are the dominant contributors to adipose tissue inflammation, forming crown-like structures in obesity (Murray et al, 2014). In rodents, recruited adipose tissue macrophages (ATMs) appear to have a proinflammatory phenotype (M1-like), whereas resident ATMs display an alternatively activated M2-like phenotype (Lumeng et al, 2007b). High fat diet (HFD) challenge induces an ATM polarization switch from the M2 state to the M1 proinflammatory state in mice (Lumeng et al, 2007a). Emerging evidence indicates that ATMs participate in the regulation of adipose tissue remodeling in a paracrine fashion during obesity, including adipogenesis and fibrosis. However, the precise underlying mechanisms governing the interaction of ATMs and adipocytes remain unclear.

Prostanoids are a class of lipid inflammatory mediators derived from arachidonic acid (AA) via the sequential enzymatic action of cyclooxygenases (COXs) and specific downstream synthases. Prostanoids, consisting of prostaglandin (PG) E$_2$, PGF$_{2\alpha}$, PGI$_2$, PGD$_2$, thromboxane A$_2$, exert diverse physiological and pathophysiological effects by binding to their specific G protein-coupled receptors (EP1-4, FP, IP, DP1-2, and TP receptors). Macrophage-specific deletion of COX-2 significantly accelerates HFD-induced obesity in mice (Pan et al, 2022), suggesting that macrophage COX-2-derived prostanoids have anti-adipogenic effects. The PGI$_2$/IP axis promotes adipocyte differentiation from preadipocytes in culture and in vivo (Rahman, 2019). PGE$_2$, mainly from macrophages, is the predominant PG produced in adipose tissues (Xu et al, 2016). EP3 global knockout mice exhibit a severely obese phenotype with increased food consumption even when fed a

[1]Department of Pharmacology, Tianjin Key Laboratory of Inflammatory Biology, The Province and Ministry Co-sponsored Collaborative Innovation Center for Medical Epigenetics, Key Laboratory of Experimental Hematology, School of Basic Medical Sciences, Tianjin Medical University, Tianjin, China. [2]Department of Bioinformatics, Tianjin Key Laboratory of Inflammation Biology, School of Basic Medical Sciences, Tianjin Medical University, Tianjin, China. [3]Department of Life Sciences and Health, University of Health and Rehabilitation Sciences, Qingdao, Shandong, China. [4]Department of Physiology, Institute of Basic Medicine, Hebei Medical University, Shijiazhuang, China. [5]These authors contributed equally: Wenlong Shang, Yinxiu Li. ✉E-mail: tiandanyang@hebmu.edu.cn; yuying@tmu.edu.cn

normal chow diet (Sanchez-Alavez et al, 2007). Suppression of the PGE$_2$-EP3 axis facilitates preadipocytes differentiation into adipocytes (Xu et al, 2016) and induces whitening of brown adipose tissues in mice (Tao et al, 2022a), suggesting a critical role of EP3 in adipogenesis. However, whether and how PGE$_2$ receptors in macrophages are involved in diet-induced obesity remains unclear.

In the present study, we observed that EP3 expression was downregulated in ATMs from patients with obesity and HFD-fed mice. Specific deletion of the EP3 receptor in macrophages exacerbates HFD-induced obesity in mice. EP3 deletion in macrophages promoted adipocyte differentiation through suppressing SPARC secretion. EP3 activation regulated SPARC expression in macrophages by suppressing the PKA/Sp1/Dnmt1/3a pathway. Pharmacological activation of EP3 receptor alleviates HFD-induced obesity in mice. Thus, our results demonstrated that EP3 may be a potential therapeutic target for diet-induced obesity.

# Results

## EP3 was downregulated in adipose tissue macrophages from HFD-induced mice and patients with obesity

To investigate the potential role of inflammatory PGs in regulating macrophage function during obesity, we first analyzed the alterations in PG receptor expression in macrophages using single nucleus (sNuc) sequencing data from human adipose tissue (Emont et al, 2022). Notably, EP3 was uniquely downregulated among nine PG receptors in ATMs from patients with obesity with a body mass index (BMI) range of 30–40, compared to individuals with the BMI range of 20–30 (Fig. 1A). Consistently, the elevation of PGE$_2$ production in the epididymal adipose tissue of HFD-fed mice (Fig. 1B), along with decreased mRNA expression of the EP3 receptor in macrophages from the epididymal adipose tissue of HFD-fed mice was observed compared with normal chow diet-fed mice (Fig. 1C). Among the three EP3 isoforms, EP3α predominated within bone marrow-derived macrophages (BMDMs). Notably, all EP3 isoforms were markedly downregulated in ATMs from HFD-fed mice and BMDMs exposed to palmitate (Fig. 1D,E). Pparγ suppresses EP3 expression in macrophages through inhibition of NF-κB (Sui et al, 2014). Similarly, we observed, PA inhibited EP3 expression in BMDMs by Pparγ/NF-κB pathway, silencing Pparγ attenuated the EP3 downregulation in PA-treated BMDMs (Appendix Fig. S1A–D). In the BMDM/3T3-L1 coculture system (Fig. 1F), PGE$_2$ analogue and EP3 agonist sulprostone pretreated BMDMs substantially inhibited 3T3-L1 differentiation toward adipocytes, whereas EP3 inhibitor L-798106 pretreated BMDMs promoted 3T3-L1 differentiation (Fig. 1G,H). Accordingly, the adipogenic markers C/EBPα, Fabp4, and Pparγ were significantly downregulated in the 3T3-L1 preadipocytes cocultured with sulprostone pretreated BMDMs but upregulated in those cocultured with L-798106 pretreated BMDMs (Fig. 1I–M). In BMDM/3T3-L1 coculture system, sulprostone-pretreated BMDMs inhibited 3T3-L1 differentiation toward adipocytes, the inhibitory effect was abrogated in the EP3$^{-/-}$ BMDMs or by EP3 inhibitor L-798106 (Appendix Fig. S2A–D). These observations suggest that the PGE$_2$/EP3 axis in macrophages may be involved in adipogenesis.

## EP3 deficiency in macrophages exacerbated HFD-induced obesity in mice

To examine whether macrophage EP3 plays an important role in HFD-induced obesity, macrophage-specific EP3-deficient mice were generated by crossing EP3$^{F/F}$ mice (control mice) with LysM$^{Cre}$ transgenic mice. As expected, EP3 was efficiently ablated in the BMDMs from EP3$^{F/F}$LysM$^{Cre}$ mice (Appendix Fig. S3A). As anticipated(Zhang et al, 2023), PA enhanced the M1 polarization and decreased M2 polarization in macrophages (Appendix Fig. S3B–E). However, EP3 deletion did not significantly influence macrophage polarization with and without PA treatment (Appendix Fig. S3B–E). EP3$^{F/F}$LysM$^{Cre}$ mice fed a 16 week-normal chow diet regimen had slightly increased body weights (Appendix Fig. S4A), but maintained similar glucose tolerance, and insulin tolerance as detected using glucose tolerance test (GTT) and insulin tolerance test (ITT) (Appendix Fig. S4B–E) and adipose tissue and liver mass (Appendix Fig. S4F), as compared with EP3$^{F/F}$ mice. Enlarged adipocyte size in epididymal and inguinal WAT (Appendix Fig. S4G–K) and increased expression of fatty acid synthesis and adipogenesis genes (Appendix Fig. S4L) were observed in EP3$^{F/F}$LysM$^{Cre}$ mice fed a normal chow diet. However, after 16 week-HFD challenge, macrophage-specific EP3-deficient mice gained more weight than EP3$^{F/F}$ mice, with similar food intake between the two groups (Fig. 2A,B). Oxygen consumption (VO$_2$) and overall energy expenditure were lower in EP3$^{F/F}$LysM$^{Cre}$ mice than in EP3$^{F/F}$ mice (Appendix Fig. S5A–C). EP3$^{F/F}$LysM$^{Cre}$ mice exhibited moderate insulin resistance, as indicated by hyperinsulinemic-euglycemic clamp test (Fig. 2C,D), enhanced adipose tissue and liver mass (Fig. 2E,F), and enlarged adipocyte size, as determined through HE staining, compared with control mice (Fig. 2G–M). Thus, EP3 deficiency in macrophages leads to adipogenesis and adipocyte enlargement in mice.

EP3 deletion in macrophages did not significantly alter the expression and production of proinflammatory cytokines and anti-inflammatory cytokines in the adipose tissues (Appendix Fig. S6A–C) and ATMs from both normal diet and HFD fed-mice (Appendix Fig. S6D,E). Likewise, no obvious changes of fibrotic area and the expression of fibrotic markers were observed in eWAT between EP3$^{F/F}$ mice and EP3$^{F/F}$LysM$^{Cre}$ mice (Appendix Fig. S6F–H). CD31 expression and vascular density were reduced in eWAT in EP3$^{F/F}$LysM$^{Cre}$ mice compared those in EP3$^{F/F}$ mice (Appendix Fig. S7A–C). Elevated expression of fatty acid synthesis and adipogenesis-related genes (Fig. 2N) was also observed in the adipose tissues of EP3$^{F/F}$LysM$^{Cre}$ mice. Furthermore, the expression of BAT differentiation markers was decreased, and the expression of WAT differentiation markers were increased in BAT of EP3$^{F/F}$LysM$^{Cre}$ mice (Fig. 2O,P), indicated BAT whitening in HFD-fed EP3$^{F/F}$LysM$^{Cre}$ mice.

## EP3 deficiency suppressed SPARC secretion in macrophages to contribute to diet-induced obesity in mice

To explore the molecular mechanism of EP3 in macrophages regulating adipocyte differentiation, secreted proteins from culture medium supernatant of palmitate-treated WT and EP3$^{-/-}$ BMDMs were analyzed by 4D label-free quantitative proteomic analysis (Fig. 3A). In total, 31 upregulated and 51 downregulated proteins released by EP3-deficient BMDMs were detected compared to

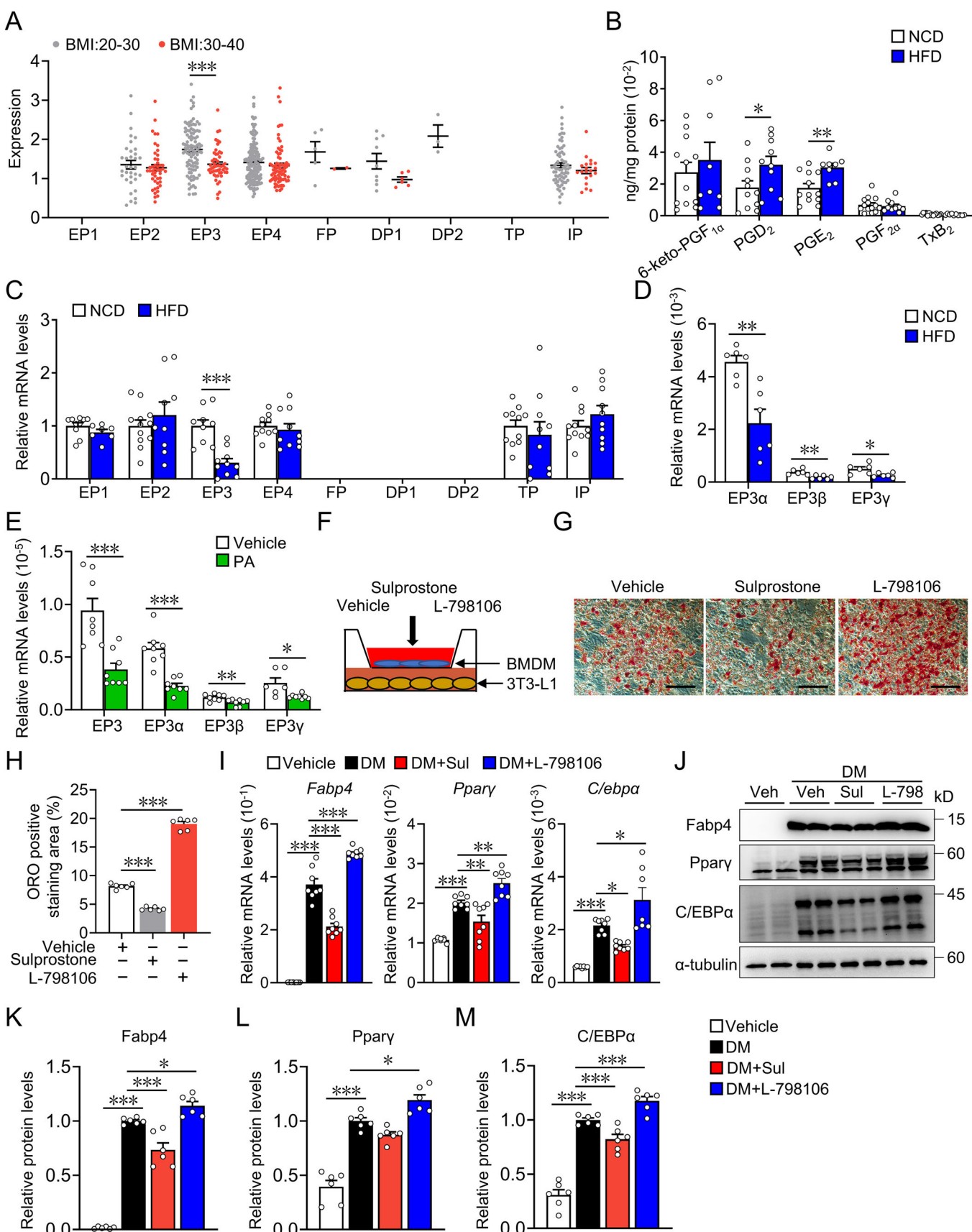

**Figure 1. Inhibition of EP3 in macrophage promoted adipogenesis.**

(A) The expression of prostaglandin (PG) receptors in macrophages of visceral adipose tissue from obese patients with the BMI range of 30–40 and individuals with the BMI range of 20–30 (GSE176171) (EP3, $P < 0.0001$). (B) PG levels in macrophages of epididymal white adipose tissue from mice fed with the normal chow diet or the HFD ($n = 8$–14) (PGD$_2$, $P = 0.0447$; PGE$_2$, $P = 0.0096$). (C) qRT-PCR analysis was used to determine mRNA expression levels of PG receptors in macrophages of epididymal white adipose tissue from mice fed with normal chow diet or HFD ($n = 7$–12) (EP3, $P = 0.0004$). (D) qRT-PCR analysis of the relative mRNA levels of EP3 isoforms in ATMs from mice fed with NCD or HFD ($n = 6$) (EP3α, $P = 0.0024$; EP3β, $P = 0.0033$; EP3γ, $P = 0.0167$). (E) qRT-PCR analysis of the relative mRNA levels of EP3 and its isoforms in BMDMs with the treatment of palmitate for 24 h ($n = 6$–8) (EP3, $P = 0.0006$; EP3α, $P = 0.0002$; EP3β, $P = 0.007$; EP3γ, $P = 0.02$). (F) Schematic of 3T3-L1 cells cocultured with BMDMs treated with vehicle, EP3 receptor agonist (sulprostone) or inhibitor (L-798106). (G) Representative images of Oil Red O staining of adipogenesis of 3T3-L1 cells cocultured with BMDMs stimulated by vehicle, EP3 receptor agonist or inhibitor (scale bar: 50 μm). (H) Quantification of Oil Red O staining in (G) ($n = 6$) (Vehicle vs Sulprostone: $P < 0.0001$; Vehicle vs L-798106: $P < 0.0001$). (I) qRT-PCR analysis of the relative mRNA levels of adipogenic gene expression in differentiated 3T3-L1 cells cultured with conditioned medium of BMDMs stimulated by vehicle, EP3 receptor agonist, or inhibitor ($n = 7$–8). (I) DM vs Vehicle, DM vs DM+Sul, DM vs DM + L-798106 in *Fabp4* graph ($P < 0.0001$; $P < 0.0001$; $P < 0.0001$), *Ppary* graph ($P < 0.0001$; $P = 0.0066$; $P = 0.0063$) and *C/ebpa* graph ($P < 0.0001$; $P = 0.0365$; $P = 0.0112$). (J) Western blot analysis of Fabp4, Ppary, C/EBPα in differentiated 3T3-L1 cells cultured with conditioned medium of BMDMs stimulated by vehicle, EP3 receptor agonist or inhibitor. (K–M) Quantification of protein levels of Fabp4 (K), Ppary (L) and C/EBPα (M) in (J) ($n = 6$). (K–M) DM vs Vehicle, DM vs DM+Sul, DM vs DM + L-798106 in (K) ($P < 0.0001$; $P = 0.0002$; $P = 0.043$), (L) ($P < 0.0001$; $P = 0.1248$; $P = 0.0135$) and (M) ($P < 0.0001$; $P < 0.0001$; $P < 0.0001$). Data information: Data represent the mean ± SEM. Data are representative of two independent experiments with biological replicates (B–E, H, I, K–M). Statistics: Mann–Whitney $U$ test (A–C, E), Unpaired Student's $t$ test (D), one-way ANOVA (H, I, K–M). (A–E, H–I, K–M), $P$ values are indicated by asterisks, with *$P < 0.05$, **$P < 0.01$, ***$P < 0.001$. Source data are available online for this figure.

control BMDMs (fold change > 2, $P < 0.05$) (Fig. 3B). Gene ontology (GO) analysis revealed notable enrichment of extracellular matrix structural constituents in these differentially secreted proteins (Fig. 3C). Among the structural constituents of the ECM, Secreted protein acidic and rich in cysteine (SPARC), an anti-adipogenic factor (Nie and Sage, 2009), was significantly decreased in the culture supernatant of EP3$^{-/-}$ BMDMs (Fig. 3D). The reduction of SPARC protein and mRNA expression in EP3$^{-/-}$ BMDMs was further verified using western blotting and qRT-PCR (Fig. 3E–G). SPARC is highly expressed in ASPC, adipocytes, endothelial, mesothelium and macrophages in both human and mouse white adipose tissue (Emont et al, 2022) (Appendix Fig. S8A,B), the expression of *Sparc* in macrophages was lower than that in adipocytes (Appendix Fig. S8C). SPARC expression was elevated in BMDMs upon EP3 agonist sulprostone treatment, which was abolished in the EP3$^{-/-}$ BMDMs or by EP3 inhibitor L-798106 (Fig. 3H,I). *Sparc* expression was consistently reduced in ATMs of EP3$^{F/F}$LysM$^{Cre}$ mice (Appendix Fig. S8D). Interestingly, *Sparc* was also downregulated in ATMs from patients with obesity and HFD-fed mice (Appendix Fig. S8E,F). Notably, knockdown of EP3α isoform, not EP3β and EP3γ, resulted in a significant reduction of *Sparc* expression in mouse BMDMs (Fig. 3J,K). Conversely, EP3α-overexpressing macrophages (Appendix Fig. S9A,B) exhibited elevated *Sparc* expression levels (Fig. 3L–N), and silencing of SPARC abolished the anti-adipogenic effect of EP3α-overexpressing macrophages on 3T3-L1 cells (Fig. 3O–T). Importantly, compared with HFD-treated control mice, those with macrophage-specific EP3α overexpression (Rosa-EP3α/LysM$^{Cre}$) demonstrated significant amelioration in HFD-induced weight gain (Fig. 4A), improvement in glucose tolerance (Fig. 4B,C), alleviation in insulin resistance (Fig. 4D,E; Appendix Fig. S9C,D), decrease in white adipose tissue mass (Fig. 4F), reduction in adipocyte size (Fig. 4G–K), and downregulation in the expression of fatty acid synthesis and adipogenesis genes in adipose tissues (Fig. 4L–P). Notably, knockdown of SPARC in Rosa-EP3α/LysM$^{Cre}$ mice (Appendix Fig. S9E) abrogated the protective effects of macrophage EP3α overexpression against obesity in mice (Fig. 4A–P). Collectively, EP3 deficiency reduces the expression and secretion of SPARC within macrophages, thereby accelerating the development of diet-induced obesity in mice.

## EP3 regulated SPARC expression in macrophages through Dnmt1/3a mediated DNA methylation

The downregulation of SPARC expression via DNA methylation is commonly observed in chronic inflammation and cancer (Nagaraju and El-Rayes, 2013). Bisulfite sequencing revealed a notable increase in DNA methylation within the Sparc promoter region in EP3-deficient macrophages (Fig. 5A–C). Furthermore, EP3 deletion in macrophages boosted the expression of DNA methyltransferases Dnmt1 and Dnmt3a, but not Dnmt3b, TET1, TET2, or TET3 (Fig. 5D–F). Treatment with the DNA methyltransferase inhibitor, azacitidine, restored SPARC expression in EP3-deficient macrophages (Fig. 5G). In addition, Dnmt1/3a knockdown effectively promoted SPARC expression in EP3-deficient macrophages (Fig. 5H). Notably, Dnmt1/3a knockdown abolished the pro-adipogenic effect of EP3-deficient macrophages on adipose precursor cells (Fig. 5I–N). Thus, EP3-deficiency in macrophages increased their pro-adipogenic capacity by modulating Sparc DNA methylation.

## Dnmt1/3a ablation attenuated the exacerbated HFD-induced obesity in macrophage-specific EP3-deficient mice

To explore whether Dnmt1/3a deletion rescued the obesity phenotype in HFD-fed EP3$^{F/F}$LysM$^{Cre}$ mice, Dnmt1/3a-floxed mice were crossed with EP3$^{F/F}$LysM$^{Cre}$ mice to generate EP3$^{F/F}$Dnmt(1,3a)$^{2F/2F}$LysM$^{Cre}$ mice (Appendix Fig. S10A–C). As expected, *Sparc* expression was significantly elevated in the Dnmt1/3a-deficient macrophages (Appendix Fig. S10D). Dnmt1/3a ablation markedly inhibited weight gain (Fig. 6A), improved glucose tolerance (Fig. 6B,C) and insulin resistance (Fig. 6D,E), and decreased the enlarged white fat mass and adipocyte size in EP3$^{F/F}$LysM$^{Cre}$ mice (Fig. 6F–K), thereby reducing the elevated expression of fatty acid synthesis and adipogenic genes in the adipose tissues of EP3$^{F/F}$LysM$^{Cre}$ mice (Fig. 6L–P).

## EP3 regulated Dnmt1/3a expression in macrophages via PKA/Sp1 pathway

Four transcription factors ($P < 10^{-5}$)-Sp1, Sp4, EGR2, and Zfp281, were identified as potentially regulating Dnmt1/3a transcription via

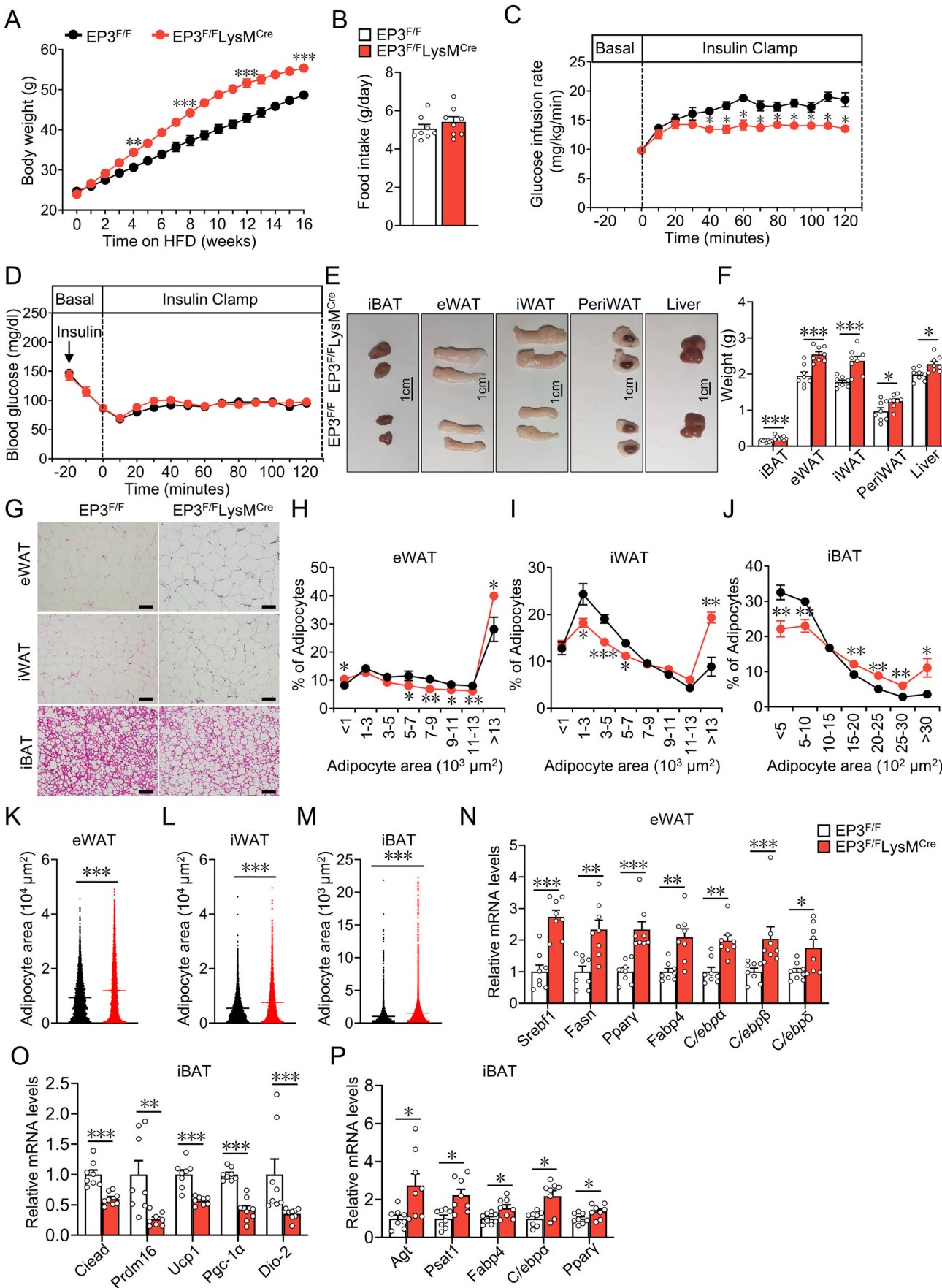

**Figure 2. Macrophage-specific deletion of EP3 aggravated HFD-induced obesity in mice.**

(A) Body weight analysis of EP3$^{F/F}$ and EP3$^{F/F}$LysM$^{Cre}$ mice fed with HFD ($n = 8$) (4 weeks, $P = 0.0045$; 8 weeks, $P = 0.0009$; 12 weeks, $P = 0.0002$; 16 weeks, $P = 0.0002$). (B) Food intake of EP3$^{F/F}$ and EP3$^{F/F}$LysM$^{Cre}$ mice fed with HFD ($n = 8$). (C, D) Hyperinsulinemic-euglycemic clamp test of EP3$^{F/F}$ and EP3$^{F/F}$LysM$^{Cre}$ mice fed with HFD ($n = 4$) (Every comparison $P = 0.0286$). (E) Representative image of iBAT, eWAT, iWAT, periWAT, and liver in EP3$^{F/F}$ and EP3$^{F/F}$LysM$^{Cre}$ mice fed with HFD. (F) The weights of adipose tissues of iBAT, eWAT, iWAT, periWAT, and liver in EP3$^{F/F}$ and EP3$^{F/F}$LysM$^{Cre}$ mice fed with HFD ($n = 8$) (iBAT, $P = 0.0008$; eWAT, $P = 0.0004$; iWAT, $P = 0.0006$; PeriWAT, $P = 0.0298$; Liver, $P = 0.0115$). (G) Representative image of hematoxylin and eosin (H&E) staining for eWAT, iWAT and iBAT from EP3$^{F/F}$ and EP3$^{F/F}$LysM$^{Cre}$ mice fed with HFD (scale bar: 100 μm). (H–M) Quantification of adipocyte area of eWAT (H, K), iWAT (I, L) and iBAT (J, M) in (G) ($n > 3358$ adipocytes measured from 6 to 8 mice in each group). (H) $<1 \times 10^3$ μm$^2$ adipocyte, $P = 0.0335$; 5–7 $\times 10^3$ μm$^2$ adipocyte, $P = 0.0491$; 7–9 $\times 10^3$ μm$^2$ adipocyte, $P = 0.0042$; 9–11 $\times 10^3$ μm$^2$ adipocyte, $P = 0.0302$; 11–13 $\times 10^3$ μm$^2$ adipocyte, $P = 0.0041$; $>13 \times 10^3$ μm$^2$ adipocyte, $P = 0.0142$. (I) 1–3 $\times 10^3$ μm$^2$ adipocyte, $P = 0.0401$; 3–5 $\times 10^3$ μm$^2$ adipocyte, $P = 0.0006$; 5–7 $\times 10^3$ μm$^2$ adipocyte, $P = 0.0205$; $>13 \times 10^3$ μm$^2$ adipocyte, $P = 0.0012$. (J) $<5 \times 10^2$ μm$^2$ adipocyte, $P = 0.007$; 5–10 $\times 10^2$ μm$^2$ adipocyte, $P = 0.0045$; 15–20 $\times 10^2$ μm$^2$ adipocyte, $P = 0.0026$; 20–25 $\times 10^2$ μm$^2$ adipocyte, $P = 0.0027$; 25–30 $\times 10^2$ μm$^2$ adipocyte, $P = 0.0082$; $>30 \times 10^2$ μm$^2$ adipocyte, $P = 0.0206$. (K) eWAT, $P < 0.0001$. (L) iWAT, $P < 0.0001$. (M) iBAT, $P < 0.0001$. (N) qRT-PCR analysis of the relative mRNA levels of fatty acid synthesis and adipogenesis genes in eWAT from EP3$^{F/F}$ and EP3$^{F/F}$LysM$^{Cre}$ mice ($n = 8$) (Srebf1, $P = 0.0006$; Fasn, $P = 0.003$; Ppary, $P = 0.0002$; Fabp4, $P = 0.0019$; C/ebpa, $P = 0.0011$; C/ebpβ, $P = 0.0003$; C/ebpδ, $P = 0.0379$). (O, P) qRT-PCR analysis of the relative mRNA levels of BAT differentiation markers (O) and WAT differentiation markers (P) in iBAT from EP3$^{F/F}$ and EP3$^{F/F}$LysM$^{Cre}$ mice ($n = 8$). (Ciead, $P = 0.0002$; Prdm16, $P = 0.0011$; Ucp1, $P = 0.0003$; Pgc-1a, $P = 0.0002$; Dio-2, $P = 0.0002$; Agt, $P = 0.0379$; Psat1, $P = 0.0104$; Fabp4, $P = 0.0281$; C/ebpa, $P = 0.0207$; Ppary, $P = 0.0379$). Data information: Data represent the mean ± SEM. Data are pooled from two independent experiments with biological replicates (A, F, H–P). Statistics: Mann–Whitney $U$ test (A, C, I, K–P). Unpaired Student's $t$ test (F, H, J). (A, C, F, H–P) $P$ values are indicated by asterisks, with *$P < 0.05$, **$P < 0.01$, ***$P < 0.001$. Source data are available online for this figure.

motif-binding site analysis in the promoter region (Fig. 7A). Silencing Sp1, but not Sp4, EGR2, or Zfp281, abolished the enhanced expression of Dnmt1 and Dnmt3a in EP3-deficient macrophages (Fig. 7B,C and Appendix Fig. S11A–D). Sp1 binding motif appeared in the promoter regions of both Dnmt1 and Dnmt3a (Fig. 7D), as reported previously(Jinawath et al, 2005; Kishikawa et al, 2002). Luciferase reporter assays showed that the EP3 agonist suppressed Sp1-mediated Dnmt1/3a transcription (Fig. 7E,F). Additionally, both the Sp1 inhibitor Mithramycin A and the PKA inhibitor H-89 effectively attenuated the phosphorylation levels of Sp1 and expression of Dnmt1/3a and concomitantly promoted SPARC protein expression in EP3-deficient macrophages (Fig. 7G,H). Finally, G$_{\alpha i}$ inhibitor Pertussis toxin augmented Dnmt1/3a expression while inhibiting SPARC expression in EP3α overexpressed macrophages (Fig. 7I). Thus, EP3 activation upregulates SPARC expression in macrophages by inhibiting the PKA/Sp1/Dnmt1/3a pathway (Fig. 7J).

## EP3 receptor activation ameliorated HFD-induced obesity in mice

To clarify whether targeting EP3 has therapeutic potential in obesity, the EP3 agonist sulprostone was administered to HFD-fed mice for an additional 5 weeks (Fig. 8A). Compared with vehicle-treated mice, sulprostone-treated mice showed significant weight loss (Fig. 8B), improved glucose tolerance (Fig. 8C,D) and insulin resistance (Fig. 8E,F), decreased white fat mass (Fig. 8G), reduced adipocyte size (Fig. 8H–L), and downregulated expression of fatty acid synthesis and adipogenesis genes in the adipose tissues (Appendix Fig. S12A–G). Furthermore, sulprostone-treated mice exhibited downregulated Dnmt1/3a expression (Fig. 8M,N; Appendix Fig. S12H,I) and increased SPARC expression (Fig. 8O,P) in ATMs.

## Discussion

Infiltrated macrophages play a critical role in adipose tissue remodeling processes, such as adipogenesis and angiogenesis. In this study, we observed that macrophages suppress white adipogenesis in vitro and in mice by secreting the anti-adipogenic factor SPARC. Activation of the PGE$_2$-EP3 axis promotes SPARC expression in macrophages by decreasing DNA methylation. EP3 agonist treatment mitigates diet-induced obesity in HFD-fed mice via SPARC. EP3 expression was downregulated in ATMs from patients with obesity and HFD-fed mice. Therefore, reactivation of EP3-mediated signaling may be a promising strategy for treating obesity.

Obesity is a metabolic state generated by adipose tissue expansion, which occurs through both adipocyte hyperplasia (increase in adipocyte number) and hypertrophy (increase in adipocyte size) (Sakers et al, 2022). Theoretically, hyperplastic growth is associated with smaller adipocytes accompanied by a better insulin sensitivity, lower inflammation level and less ectopic lipid accumulation (Nishimura et al, 2008; Strissel et al, 2007; Weyer et al, 2000). Adipocyte hypertrophy leads to dysregulated adipose tissues in obesity characterized by a proinflammatory profile and enhanced insulin resistance (Ghaben and Scherer, 2019; Gustafson et al, 2009). In our study, EP3 deficiency in macrophages promotes adipocytes hyperplasia and hypertrophy in HFD-challenged mice. Likewise, deletion of some metabolic-related genes, such as FAS (Lodhi et al, 2012), Abca1 (Cuffe et al, 2018), regulates HFD-induced obesity in mice via interrupting both adipocytes hypertrophy and differentiation.

Obesity is a state of chronic low-grade systemic inflammation with massive macrophage infiltration into various metabolic tissues, including the adipose tissue. Interactions between macrophages and adipocytes are critical for adipose tissue remodeling. Macrophages eliminate dead adipocytes via exophagy to maintain adipose tissue homeostasis (Haka et al, 2016). Upon cold stimulation, ATMs promote WAT beiging by secreting Slit3 cytokine and increasing sympathetic activation (Wang et al, 2021). Proinflammatory ATMs enhance expression of matrix metalloproteinases in adipocytes and induce adipose tissue progenitor cell differentiation toward fibroblast-like cells, subsequently facilitate tissue fibrosis (Gao and Bing, 2011; O'Hara et al, 2009) through IL-1β and TGF1β cytokines (Bourlier et al, 2012). TIM4$^+$ATMs regulate adipocyte expansion and lipid storage in diet-induced obese mice by secreting PDGFcc, and blocking ATM-derived PDGFcc with neutralizing antibodies increases energy expenditure via thermogenesis in brown fat (Cox et al, 2021). In addition, CD11c$^+$ ATMs communicate with adipocytes via the

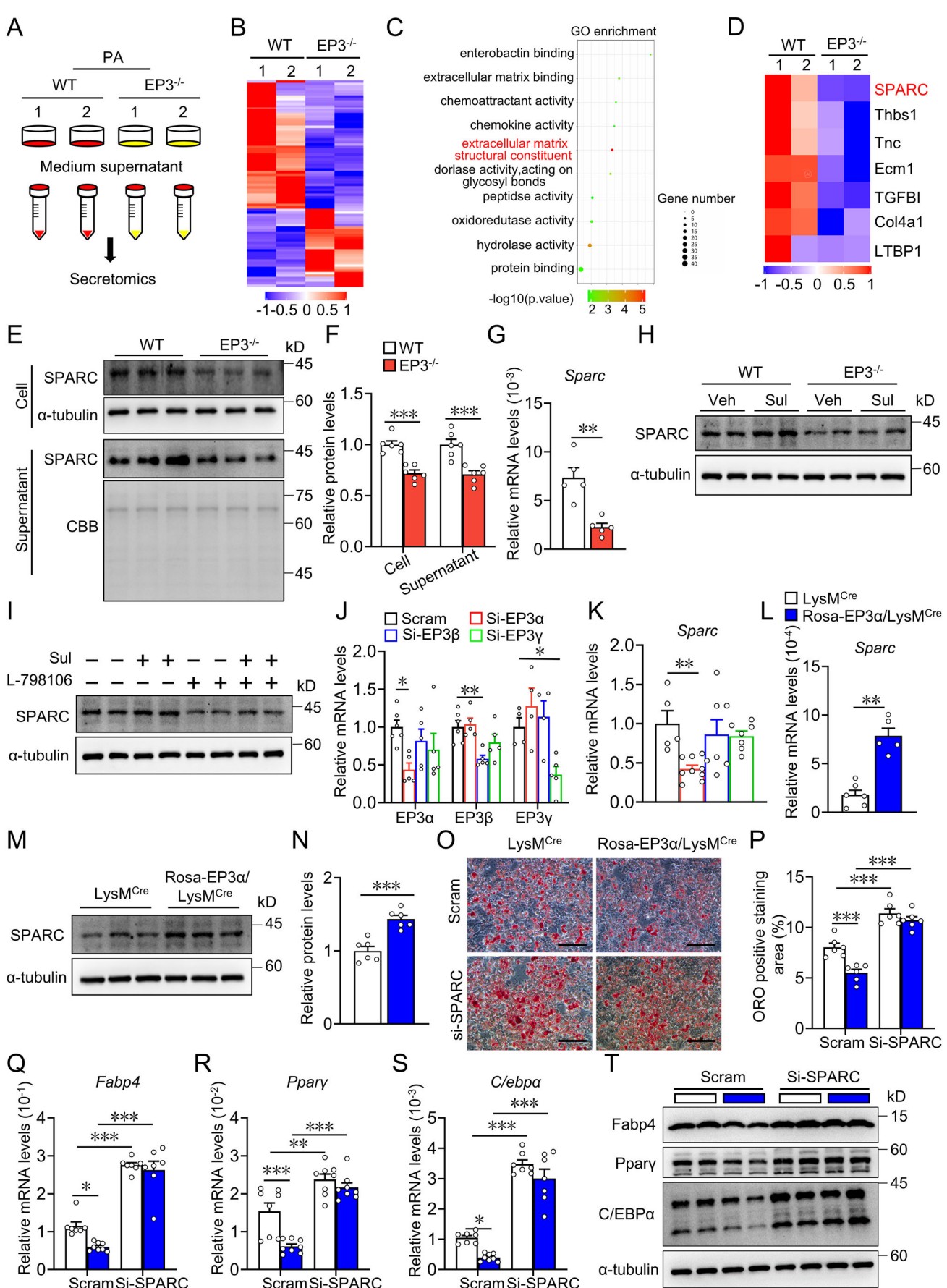

**Figure 3. EP3 deficiency promoted adipocyte differentiation by reducing SPARC secretion in macrophage.**

(A) Schematic of medium supernatant of BMDMs from WT and EP3$^{-/-}$ mice were collected for 4D label-free quantitative proteomic analysis after PA treatment for 36 h. (B) Heatmap of secretory proteins in medium supernatant of BMDMs from WT and EP3$^{-/-}$ mice with treatment of palmitate detected by the 4D label-free quantitative proteomic analysis. (C) Top 10 enriched gene ontology (GO) terms in the molecular function of up- and downregulated proteins in medium supernatant of EP3$^{-/-}$ BMDMs with treatment of palmitate compared to that of WT BMDMs. (D) Heatmap of protein expression levels of extracellular matrix structural constituent pathway-related genes (SPARC, Thbs1, Tnc, Ecm1, TGBFI, Col4a1, and LTBP1) in WT and EP3$^{-/-}$ BMDMs with treatment of palmitate. (E) Western blot analysis of SPARC in WT and EP3$^{-/-}$ BMDMs with treatment of palmitate (up) and medium supernatant (down) with PA treatment, Coomassie brilliant blue (CBB) staining as loading controls. (F) Quantification of SPARC protein levels in (E) ($n = 6$) (Cell, $P = 0.0001$; Supernatant, $P = 0.0009$). (G) qRT-PCR analysis of the relative mRNA levels of Sparc in WT and EP3$^{-/-}$ BMDMs with treatment of palmitate ($n = 5$) ($P = 0.0079$). (H) Western blot analysis of SPARC expression in sulprostone-treated WT and EP3$^{-/-}$ BMDMs. (I) Effect of L-798106 on SPARC expression in sulprostone-treated BMDMs. (J–K) qRT-PCR analysis of the relative mRNA levels of EP3 isoforms (J) ($n = 4$-6), Sparc (K) ($n = 5$-9) in BMDMs transfected with EP3 isoforms siRNA. (EP3α Scram vs Si-EP3α, $P = 0.0289$; EP3β Scram vs Si-EP3β, $P = 0.0048$; EP3γ Scram vs Si- EP3γ, $P = 0.0487$; Sparc Scram vs Si-EP3α, $P = 0.0081$). (L) qRT-PCR analysis of the relative mRNA levels of Sparc in BMDMs from LysM$^{Cre}$ and Rosa-EP3α/LysM$^{Cre}$ mice ($n = 5$-6) ($P = 0.0043$). (M) Western blot analysis of SPARC expression in BMDMs from LysM$^{Cre}$ and Rosa-EP3α/LysM$^{Cre}$ mice. (N) Quantification of protein levels in (M) ($n = 6$) ($P = 0.0003$). (O) Representative images of Oil Red O staining of differentiated 3T3-L1 cells cocultured with BMDMs from LysM$^{Cre}$ and Rosa-EP3α/LysM$^{Cre}$ mice transfected with SPARC siRNA (scale bar: 50 μm). (P) Quantification of Oil Red O staining in (O) ($n = 6$) (Scram LysM$^{Cre}$ vs Scram Rosa-EP3α/LysM$^{Cre}$, $P = 0.001$; Scram LysM$^{Cre}$ vs Si-SPARC LysM$^{Cre}$, $P < 0.0001$; Scram Rosa-EP3α/LysM$^{Cre}$ vs Si-SPARC Rosa-EP3α/LysM$^{Cre}$, $P < 0.0001$). (Q–S) qRT-PCR analysis of the relative mRNA levels of Fabp4 (Q, $n = 7$-8) and Ppary (R, $n = 8$) and C/ebpα (S, $n = 7$-8) in differentiated 3T3-L1 cells cultured with conditioned medium of BMDMs from LysM$^{Cre}$ and Rosa-EP3α/LysM$^{Cre}$ mice transfected with SPARC siRNA. (Q–S) Scram LysM$^{Cre}$ vs Scram Rosa-EP3α/LysM$^{Cre}$, Scram LysM$^{Cre}$ vs Si-SPARC LysM$^{Cre}$, Scram Rosa-EP3α/LysM$^{Cre}$ vs Si-SPARC Rosa-EP3α/LysM$^{Cre}$ in (Q) ($P = 0.0242$; $P < 0.0001$; $P < 0.0001$), (R) ($P = 0.0008$; $P = 0.0023$; $P < 0.0001$), (S) ($P = 0.0427$; $P < 0.0001$; $P < 0.0001$). (T) Western blot analysis of Fabp4, Ppary, C/EBPα in differentiated 3T3-L1 cells cultured with conditioned medium of BMDMs from LysM$^{Cre}$ and Rosa-EP3α/LysM$^{Cre}$ mice transfected with SPARC siRNA. Data information: Data represent the mean ± SEM. Data are representative of two independent experiments with biological replicates (F, G, J–L, N, P–S). Statistics: Unpaired Student's t test (F, N), Mann–Whitney U test (G, L), one-way ANOVA (J, K), two-way ANOVA (P–S). (F, G, J–L, N, P–S) P values are indicated by asterisks, with *$P < 0.05$, **$P < 0.01$, ***$P < 0.001$. Source data are available online for this figure.

growth/differentiation factor 3 (GDF3)-ALK7 pathway, thus promoting adipogenesis (Bu et al, 2018). However, senescent CD9$^+$ ATMs exhibit impaired immune function and inhibit the adipogenic differentiation of stromal cells by secreting osteopontin in HFD-challenged mice (Rabhi et al, 2022). Here, we report a novel macrophage-adipocyte interaction counters the development of obesity through EP3-governed macrophage secretion of SPARC, an anti-adipogenic factor, to suppress pre-adipocyte differentiation toward to mature adipocytes.

SPARC is an extracellular glycoprotein that is expressed in various tissues and has pleiotropic functions. In adipose tissue, SPARC is mainly expressed in stromal cells and adipocytes and is upregulated in patients with obesity and animals (Tartare-Deckert et al, 2001; Tseng and Kolonin, 2016). SPARC deficiency leads to adiposity in mice with increased adipocyte size and number (Bradshaw et al, 2003), probably through disrupting ECM deposition and inhibiting Wnt/β-catenin-mediated pre-adipocyte differentiation and proliferation (Nie et al, 2011; Nie and Sage, 2009). Interestingly, SPARC also improves fat oxidation to limit lipid deposition (Mukherjee et al, 2020). SPARC is expressed in skeletal muscle progenitor cells (Jorgensen et al, 2009) and may regulate skeletal muscle proliferation and differentiation (Nakamura et al, 2012). Adenovirus-mediated SPARC inhibition results in enhanced intramuscular adipose tissue formation in aged animals (Mathes et al, 2021). Additionally, SPARC is detected in fatty liver tissues, and its downregulation facilitates hepatic lipogenesis in mice (A et al, 2021) by modulating AMPK activity via direct protein–protein interaction (Song et al, 2010). We observed reduced Sparc expression in the ATMs of myeloid EP3 deficient mice along with increased adipogenesis and deletion of SPARC in macrophages abrogated the anti-adipogenic effect of EP3, further confirming the critical role of SPARC in adipogenesis and metabolic health.

DNA methylation, an important epigenetic regulation, is a biochemical process in which methyl provided from S-adenosylmethionine is added to the 5′ position (C) of cytosine

or the 6′ position (a) of the adenine nitrogen ring by DNA methyltransferases (Dnmt1, Dnmt3a, Dnmt3b). Abnormal DNA methylations of key metabolic-related genes, such as Leptin (Houde et al, 2015), adiponectin (Houde et al, 2015), PGC1α (Brons et al, 2010), IGF-2 (Perkins et al, 2012), appetite-related genes-(POMC (Crujeiras et al, 2013), NPY (Crujeiras et al, 2013)), are frequently implicated in the development of obesity and insulin resistance. We observed an upregulated methylation level in the Sparc promoter region in obese mice, leading to significantly decreased Sparc expression in ATMs. Consistently, Sparc reduction by dysregulated methylation in the promoter region has been reported in other chronic diseases such as age-dependent disc degeneration (Tajerian et al, 2011), Sjogren's syndrome (Feldt et al, 2022), and colon cancer (Cheetham et al, 2008).

As a prominent inflammatory prostanoid, PGE$_2$ plays an important role in the physiology and pathophysiology (Civelek and Ozen, 2022). PGE$_2$ is synthesized sequentially by cyclooxygenases (COXs) and prostaglandin E synthase (PGES) enzymes. There are three PGES isoforms—microsomal PGES-1 (mPGES-1), mPGES-2, and cytosolic PGES (cPGES). PGE$_2$ can be generated by most of cells in adipose tissues since PGESs are expressed in most cells. However, in the inflamed tissues, PGE$_2$ is dominantly produced in macrophages through inducible COX-2 and mPGES-1 (Martin-Vazquez et al, 2023). In HFD-induced obese mouse model, COX-2 is upregulated in adipose tissues (Garcia-Alonso et al, 2016; Hsieh et al, 2009), along with increased PGE$_2$ production. Exogenous PGE$_2$ treatment markedly suppresses fibrogenic and inflammatory gene expression but increases brown marker expression in human WAT implants (Garcia-Alonso et al, 2016). PGE$_2$ negatively regulates adipogenesis in mouse embryonic fibroblasts and 3T3-L1 cells (Inazumi et al, 2011; Wang et al, 2022). PGE$_2$ plays a vital role in the accumulation of adipocytes and development of obesity by acting on its four receptors, EP1, EP2, EP3, and EP4 (Xu et al, 2016). The PGE$_2$-EP2 axis mediates diet-induced obesity by activating melanin-concentrating hormone (MCH) neurons in the hypothalamus (Fang et al, 2023). EP4

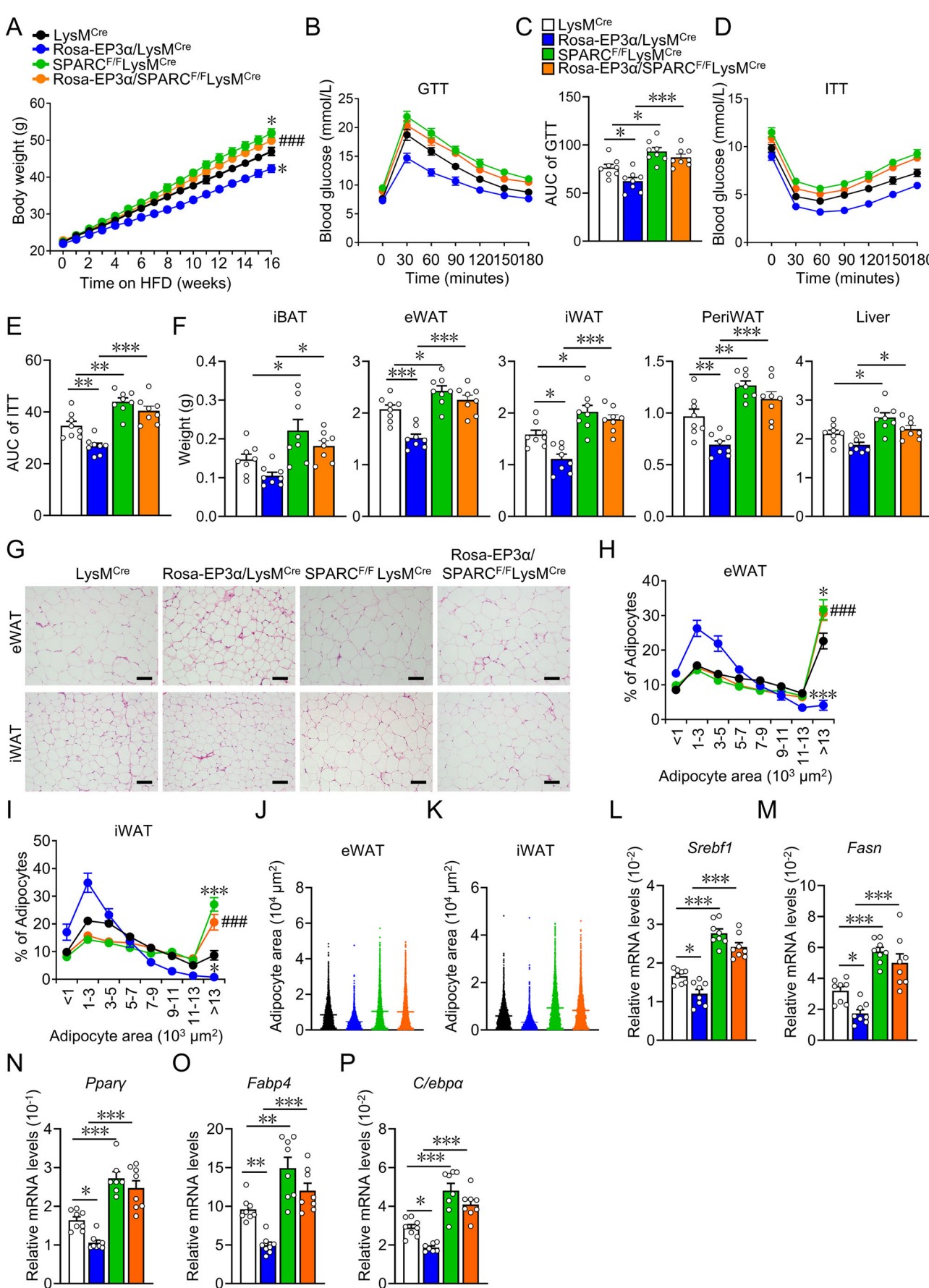

◄

**Figure 4.   SPARC deletion attenuated EP3 overexpression-alleviated HFD-induced obesity in mice.**

(A) Body weight analysis of LysM$^{Cre}$, Rosa-EP3α/LysM$^{Cre}$, SPARC$^{F/F}$LysM$^{Cre}$, and Rosa-EP3α/SPARC$^{F/F}$LysM$^{Cre}$ mice fed with HFD ($n = 8$) (16 weeks LysM$^{Cre}$ vs Rosa-EP3α/LysM$^{Cre}$, $P = 0.0224$; 16 weeks LysM$^{Cre}$ vs SPARC$^{F/F}$LysM$^{Cre}$, $P = 0.0163$; 16 weeks Rosa-EP3α/LysM$^{Cre}$ vs Rosa-EP3α/SPARC$^{F/F}$LysM$^{Cre}$, $P = 0.0001$). (B) GTT in LysM$^{Cre}$, Rosa-EP3α/LysM$^{Cre}$, SPARC$^{F/F}$LysM$^{Cre}$, and Rosa-EP3α/SPARC$^{F/F}$LysM$^{Cre}$ mice fed with HFD ($n = 8$). (C) AUC of the GTT described in (B) ($n = 8$) (LysM$^{Cre}$ vs Rosa-EP3α/LysM$^{Cre}$, $P = 0.0463$; LysM$^{Cre}$ vs SPARC$^{F/F}$LysM$^{Cre}$, $P = 0.0128$; Rosa-EP3α/LysM$^{Cre}$ vs Rosa-EP3α/SPARC$^{F/F}$LysM$^{Cre}$, $P = 0.0002$). (D) ITT in LysM$^{Cre}$, Rosa-EP3α/LysM$^{Cre}$, SPARC$^{F/F}$LysM$^{Cre}$, and Rosa-EP3α/SPARC$^{F/F}$LysM$^{Cre}$ mice fed with HFD ($n = 8$). (E) AUC of the ITT described in (D) ($n = 8$) (LysM$^{Cre}$ vs Rosa-EP3α/LysM$^{Cre}$, $P = 0.0063$; LysM$^{Cre}$ vs SPARC$^{F/F}$LysM$^{Cre}$, $P = 0.0014$; Rosa-EP3α/LysM$^{Cre}$ vs Rosa-EP3α/SPARC$^{F/F}$LysM$^{Cre}$, $P < 0.0001$). (F) The weights of adipose tissues of iBAT, eWAT, periWAT, and liver in LysM$^{Cre}$, Rosa-EP3α/LysM$^{Cre}$, SPARC$^{F/F}$LysM$^{Cre}$, and Rosa-EP3α/SPARC$^{F/F}$LysM$^{Cre}$ mice fed with HFD ($n = 8$). (F) LysM$^{Cre}$ vs Rosa-EP3α/LysM$^{Cre}$, LysM$^{Cre}$ vs SPARC$^{F/F}$LysM$^{Cre}$, Rosa-EP3α/LysM$^{Cre}$ vs Rosa-EP3α/SPARC$^{F/F}$LysM$^{Cre}$ in iBAT graph ($P = 0.3625$; $P = 0.0303$; $P = 0.0244$), eWAT graph ($P = 0.0006$; $P = 0.0369$; $P < 0.0001$), iWAT graph ($P = 0.0123$; $P = 0.0232$; $P < 0.0001$), PeriWAT graph ($P = 0.0094$; $P = 0.0045$; $P < 0.0001$), Liver graph ($P = 0.1386$; $P = 0.0183$; $P = 0.0203$). (G) Representative image of H&E staining for eWAT and iWAT from LysM$^{Cre}$, Rosa-EP3α/LysM$^{Cre}$, SPARC$^{F/F}$LysM$^{Cre}$, and Rosa-EP3α/SPARC$^{F/F}$LysM$^{Cre}$ mice fed with HFD (scale bar: 100 μm). (H–K) Quantification of adipocyte area of eWAT (H, J) and iWAT (I to K) in (G) ($n > 3757$ adipocytes measured from 6–7 mice in each group). (H, I) >13 × 10$^3$ μm$^2$ adipocyte LysM$^{Cre}$ vs > Rosa-EP3α/LysM$^{Cre}$, > 13×10$^3$ μm$^2$ adipocyte LysM$^{Cre}$ vs > SPARC$^{F/F}$LysM$^{Cre}$, >13 × 10$^3$ μm$^2$ adipocyte Rosa-EP3α/LysM$^{Cre}$ vs > Rosa-EP3α/SPARC$^{F/F}$LysM$^{Cre}$ in (H) ($P < 0.0001$; $P = 0.0409$; $P < 0.0001$), (I) ($P = 0.0403$; $P < 0.0001$; $P < 0.0001$). (L–P) qRT-PCR analysis of the relative mRNA levels of fatty acid synthesis and adipogenesis genes in eWAT from LysM$^{Cre}$, Rosa-EP3α/LysM$^{Cre}$, SPARC$^{F/F}$LysM$^{Cre}$, and Rosa-EP3α/SPARC$^{F/F}$LysM$^{Cre}$ mice ($n = 7$–8). (L–P) LysM$^{Cre}$ vs Rosa-EP3α/LysM$^{Cre}$, LysM$^{Cre}$ vs SPARC$^{F/F}$LysM$^{Cre}$, EP3α/LysM$^{Cre}$ vs Rosa-EP3α/SPARC$^{F/F}$LysM$^{Cre}$ in (L) ($P = 0.0189$; $P < 0.0001$; $P < 0.0001$), (M) ($P = 0.0391$; $P = 0.0002$; $P < 0.0001$), (N) ($P = 0.0259$; $P < 0.0001$; $P < 0.0001$), (O) ($P = 0.0066$; $P = 0.0016$; $P < 0.0001$), (P) ($P = 0.0365$; $P < 0.0001$; $P < 0.0001$). Data information: Data represent the mean ± SEM. Data are pooled from two independent experiments with biological replicates (A–F, H–P). Statistics: two-way ANOVA (A, C, E, F, H, I, L–P). (A, C, E, F, H, I, L–P) $P$ values are indicated by asterisks, with *$P < 0.05$, **$P < 0.01$, ***$P < 0.001$ vs. LysM$^{Cre}$, ###$P < 0.001$ vs. Rosa-EP3α/LysM$^{Cre}$. Source data are available online for this figure.

activation in microglia also promotes diet-induced obesity by reducing contact with pro-opiomelanocortin neurons (Niraula et al, 2023). Paradoxically, EP4 activation in macrophages ameliorates metabolic disorders, such as obesity and insulin resistance, in HFD-fed mice by regulating phagocytic capacity (Pan et al, 2022). In our study, no obvious differences in EP4 and EP2 expression in ATMs were observed between patients with obesity with a BMI range of 30–40 and individuals with a BMI range of 20–30. However, EP3 was the only downregulated PGE$_2$ receptor in ATMs from both patients with obesity and HFD-fed mice. Indeed, EP3 global knockout mice have a strong genetic pre-disposition towards obesity (Ceddia et al, 2016; Sanchez-Alavez et al, 2007). Brown adipocyte-specific depletion of EP3 compromises interscapular BAT formation and aggravates HFD-induced obesity and insulin resistance in mice (Tao et al, 2022a). EP3 deficient adipocytes lack PGE$_2$-evoked inhibition of isoproterenol-induced lipolysis in a cAMP-PKA-dependent manner (Ceddia et al, 2016; Xu et al, 2016). Consistently, we found that deletion of EP3 in macrophages under control of LysM Cre promoted lipogenesis of preadipocytes whereas EP3 overexpression and EP3 agonist supplementation ameliorated HFD-induced obesity and adipocyte formation. Although the LysM-Cre-based gene deletion was also detected in neutrophils (Orthgiess et al, 2016), neutrophils account for only <5% of the total infiltrated immune cells in WATs from patients with obesity and obese mice compared to nearly 50–70% macrophages (Emont et al, 2022; Massier et al, 2023; Vijay et al, 2020), indicating that the phenotype is mainly contributed by EP3-deficient macrophages. Moreover, EP3 also regulates sleep architecture and feeding behavior to prevent obesity, probably in noradrenergic neurons in the locus coeruleus (Mukai et al, 2023; Sanchez-Alavez et al, 2007). Thus, EP3 activation exerts multiple anti-obesity effects and targeting EP3 may represent a promising therapeutic approach for the treatment of obesity.

Genetic manipulation studies demonstrate EP3 deficiency leads to diet-induced obesity in mice through increase food uptake (Sanchez-Alavez et al, 2007), BAT whitening (Tao et al, 2022a) and WAT adipogenesis (Xu et al, 2016), indicating EP3 involvement of multiple organs/tissues (brain, BAT and WAT) in fat metabolism. In this study, we cannot directly role out the contribution of EP3 in

other organs/tissues rather than EP3 in macrophages by administration of EP3 agonist sulprostone. However, induction of SPARC expression, and suppression of its upstream Dnmt1 expression in macrophages were observed in sulprostone-treated animals, indicating macrophage EP3 contribution.

In summary, the activation of EP3 receptor inhibits HFD-induced obesity in mice by promoting macrophage releasing SPARC via the PKA/Sp1/Dnmt1,3a pathway, indicating that the EP3 receptor may be an attractive therapeutic target for obesity.

# Methods

**Reagents and tools table**

| Reagent/resource | Reference or source | Identifier or catalog number |
|---|---|---|
| **Experimental models** | | |
| SPARC$^{F/F}$ mice | Shanghai Model Organisms Center, Inc. | NM-CKO-2109560 |
| Dnmt (1, 3a)$^{2F/2F}$ | Sun et al, 2020 | N/A |
| EP3$^{-/-}$ mice | Tang et al, 2017 | N/A |
| EP3$^{F/F}$LysM$^{Cre}$ mice | Tang et al, 2017 | N/A |
| Rosa-EP3α | Tao et al, 2022b | N/A |
| **Recombinant DNA** | | |
| PcDNA3.1-Sp1 | Kong et al, 2020 | N/A |
| pGL3-Dnmt1 | Hanheng Biotechnology | N/A |
| pGL3-Dnmt3a | Hanheng Biotechnology | N/A |
| **Antibodies** | | |
| Rabbit anti-Dnmt1 | Cell Signaling Technology | 5032 |
| Rabbit anti-Dnmt3a | Abcam | ab188470 |
| Rabbit anti-Dnmt3b | Abcam | ab2851 |

| Reagent/resource | Reference or source | Identifier or catalog number |
|---|---|---|
| Rabbit anti-SPARC | Cell Signaling Technology | 5420 |
| Rabbit anti-SP-1 | Proteintech | 21962-1-AP |
| Rabbit anti-phospho-SP1 (T453) | Abcam | ab59257 |
| Rabbit anti-Tet1 | Abclonal | A1506 |
| Rabbit anti-Tet2 | Proteintech | 21207-1-AP |
| Rabbit anti-Tet3 | Abclonal | A7612 |
| Rabbit anti-NF-κb p65 | Cell Signaling Technology | 8242S |
| Rabbit Histone H3 | Cell Signaling Technology | 4499S |
| Rabbit anti-NOS2 | Abcam | ab15323 |
| Goat anti-YM1 | R&D Systems | AF2446 |
| HRP-conjugated Rabbit anti-Goat IgG | Abclonal | AS029 |
| HRP-conjugated secondary antibodies | Cell Signaling Technology | 7074S, 7076S |
| Mac-3 | Biolegend | 108512 |
| CD31 | Abcam | ab28364 |
| Alexa Fluor 488 conjugated secondary antibodies | Invitrogen | A11008 |
| Alexa Fluor 647 conjugated secondary antibodies | Jackson ImmunoResearch | 112-606-07 |
| Anti-CD45-APCCy7 | ebioscience | 47-0451-82 |
| Anti-F4/80-Brilliant Violet 421 | BioLegend | 123137 |
| Anti-CD11b-FITC | BioLegend | 101206 |
| **Oligonucleotides and other sequence-based reagents** | | |
| siRNA sequences | This study | Appendix Table S1 |
| RT-qPCR Primers | This study | Appendix Table S2 |
| BSP methylation Primer | This study | Appendix Table S3 |
| **Chemicals, enzymes and other reagents** | | |
| Decitabine | Sigma-Aldrich | A3656 |
| Insulin | Sigma-Aldrich | I6634 |
| Collagenase II | Sigma-Aldrich | S20133006 |
| Dexamethasone | Sigma-Aldrich | D4902 |
| 3-Isobutyl-1-methylxanthine | Sigma-Aldrich | I5879 |
| Rosiglitazone | Sigma-Aldrich | D2408 |
| D-glucose | Sigma-Aldrich | G6152 |
| H-89 | Cayman | 10010556 |
| Mithramycin A | Cayman | 11434 |
| Lipofectamine™ RNAiMAX Transfection Reagent | Invitrogen | 13778075 |
| IL-6 ELISA Kit | E-EL-M0044 | Elabscience |
| TNF-α ELISA Kit | E-EL-M3063 | Elabscience |
| MCP-1 ELISA Kit | E-EL-M3001 | Elabscience |

| Reagent/resource | Reference or source | Identifier or catalog number |
|---|---|---|
| Mouse Peripheral Blood Mononuclear Cell Isolation Kit | P5230 | Solarbio |
| Palmitate Acid | KC002 | Kunchuang Biotechnology |
| TRIzol reagent | 15596018 | Invitrogen |
| Hifair® III 1st Strand cDNA Synthesis SuperMix for qPCR (gDNA digester plus) | 11141ES60 | YeaSen Biotechnology |
| Oil Red O staining | G1015 | Servicebio |
| **Software** | | |
| ImageJ | https://imagej.net/ | |
| GraphPad Prism 8.0 | https://www.graphpad.com | |
| Signal P 4.1 | http://www.cbs.dtu.dk/services/SignalP/ | |
| TMHMM server 2.0 | http://www.cbs.dtu.dk/services/TMHMM/ | |
| Secretome P 2.0 | http://www.cbs.dtu.dk/services/SecretomeP/ | |
| DAVID | https://davidbioinformatics.nih.gov/ | |
| **Other** | | |
| FACSAriaTM II flow cytometry system | BD Biosciences | |
| TimeTOF Pro mass spectrometer | Bruker | |
| ACQUITY UPLC liquid chromatography system | Waters | |
| 5500 QTRAP mass spectrometer | AB Sciex | |
| Laser confocal ultra-high-resolution microscope | ZEISS | |
| LightCycler 480 Instrument II | Roche | |
| The metabolic chambers of the CLAMS | Columbus | |
| Transwell system | Corning | |

## Animals

Male mice aged 8–12 weeks were used in all experiments. SPARC[F/F] mice (Stock No: NM-CKO-2109560) were purchased from Shanghai Model Organisms Center, Inc. EP3[−/−] (Tang et al, 2017), EP3[F/F]LysM[Cre](Tang et al, 2017), Rosa-EP3α (Tao et al, 2022b), Dnmt (1, 3a)[2F/2F] (Sun et al, 2020) mice were bred in the laboratory. We generated Rosa-EP3α/SPARC[F/F]LysM[Cre] mice by crossing SPARC[F/F] mice with Rosa-EP3α/LysM[Cre] mice. Additionally, EP3[F/F]LysM[Cre] mice were crossed with Dnmt (1, 3a)[2F/2F] mice to produce EP3[F/F]Dnmt (1, 3a)[2F/2F]LysM[Cre] mice. All strains were maintained on a C57BL/6 genetic background in specific pathogen-free animal barrier facilities at Tianjin Medical University. Mice were housed in an environment with a controlled temperature (21–23 °C), relative humidity (45–55%), a 12:12-h day–night cycle, and with free access to food and water. For HFD studies, mice were fed an HFD (60% fat; #MD12033, Medicience, Yangzhou, Jiangsu, China) for the indicated periods. All animal experiments complied

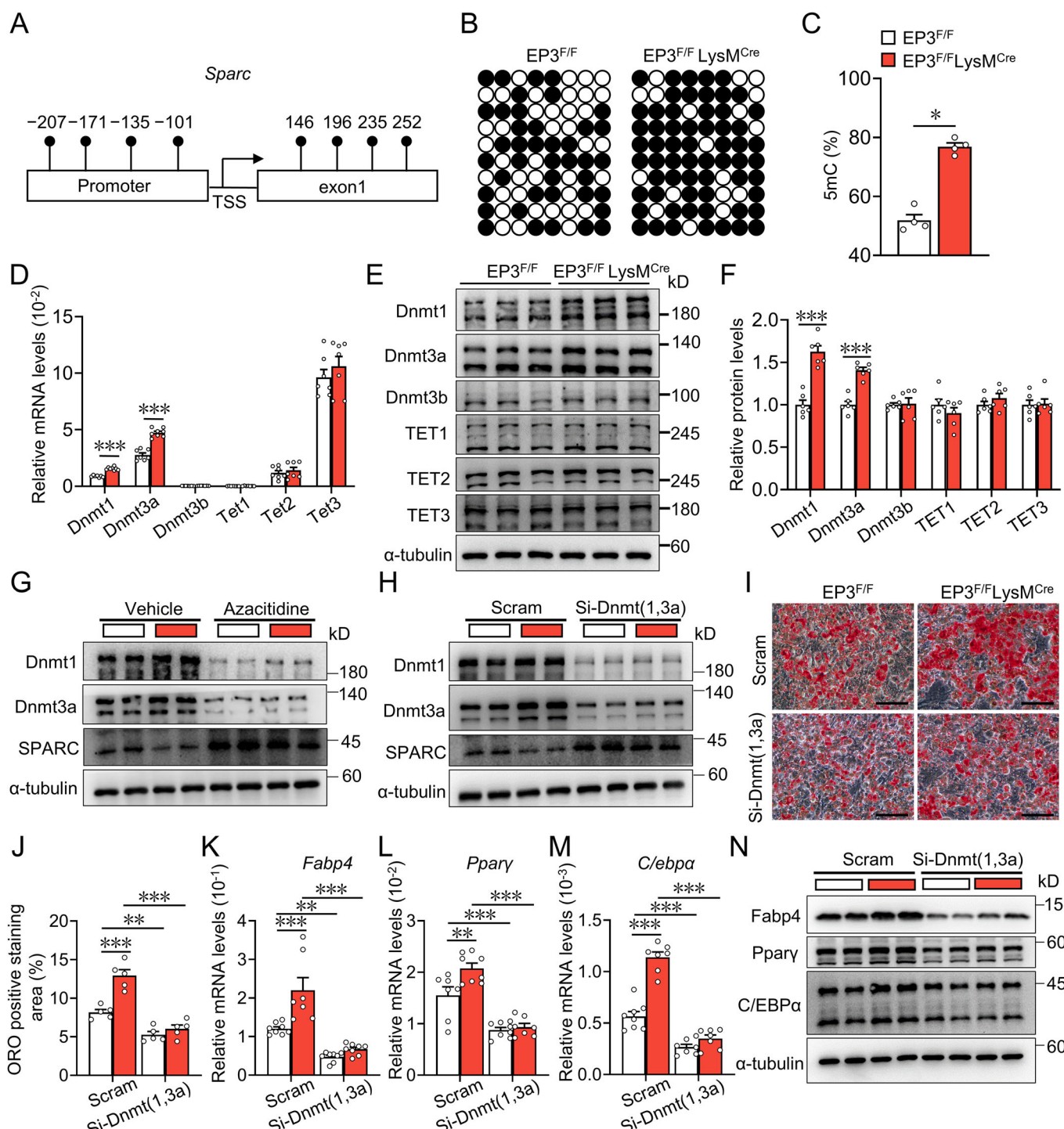

with the Guidelines of the Institutional Animal Care and Use Committee of Tianjin Medical University.

## Reagents

Decitabine (#A3656), insulin (#I6634), collagenase II (#S20133006), dexamethasone (#D4902), 3-Isobutyl-1-methylxanthine (#I5879), rosiglitazone (#D2408), and D-glucose (#G6152) were purchased from Sigma-Aldrich (St. Louis, MO, USA). H-89 (#10010556) and Mithramycin A (#11434) were purchased from the Cayman Chemical Company (Ann Arbor, MI, USA). Palmitate acid and its solvent were purchased from Kunchuang Biotechnology (Xi'an, Shaanxi, China).

## GTT and ITT

For the glucose tolerance test (GTT), mice were fasted overnight for 16 h with free access to water and subsequently injected

**Figure 5.   EP3 deletion promoted DNA methylation level of SPARC via increasing the levels of Dnmt1/3a.**

(A) The Sparc promoter region with specific CpG sites. TSS, transcription start site. (B) DNA methylation status at Sparc promoter of BMDMs from EP3$^{F/F}$ and EP3$^{F/F}$LysM$^{Cre}$ mice detected using bisulfite sequencing. Filled circles, methylated CpGs; open circles, unmethylated CpGs. (C) Quantification of the average 5mC level in (B) ($n = 4$) ($P = 0.0286$). (D) qRT-PCR analysis of the relative mRNA levels of *Dnmt1, Dnmt3a, Dnmt3b, Tet1, Tet2 and Tet3* in BMDMs from EP3$^{F/F}$ and EP3$^{F/F}$LysM$^{Cre}$ mice ($n = 7–8$) (*Dnmt1*, $P < 0.0001$; *Dnmt3a*, $P < 0.0001$). (E) Western blot analysis of Dnmt1, Dnmt3a, Dnmt3b, TET1, TET2, and TET3 in BMDMs from EP3$^{F/F}$ and EP3$^{F/F}$LysM$^{Cre}$ mice. (F) Quantification of protein levels in (E) ($n = 6$) (Dnmt1, $P < 0.0001$; Dnmt3a, $P < 0.0001$). (G) Western blot analysis of Dnmt1, Dnmt3a, SPARC in BMDMs from EP3$^{F/F}$ and EP3$^{F/F}$LysM$^{Cre}$ mice after treatment with DNA methyltransferase inhibitor (Azacitidine). (H) Western blot analysis of Dnmt1, Dnmt3a, SPARC in BMDMs from EP3$^{F/F}$ and EP3$^{F/F}$LysM$^{Cre}$ mice transfected with Dnmt1 and Dnmt3a siRNA. (I) Representative images of Oil Red O staining of differentiated 3T3-L1 cells cocultured with BMDMs from EP3$^{F/F}$ and EP3$^{F/F}$LysM$^{Cre}$ mice transfected with Dnmt1 and Dnmt3a siRNA (scale bar: 50 μm). (J) Quantification of Oil Red O staining in (I) ($n = 5$) (Scram EP3$^{F/F}$ vs Scram EP3$^{F/F}$LysM$^{Cre}$, $P < 0.0001$; Scram EP3$^{F/F}$ vs Si- Dnmt (1,3a) EP3$^{F/F}$, $P = 0.0072$; Scram EP3$^{F/F}$LysM$^{Cre}$ vs Si- Dnmt (1,3a) EP3$^{F/F}$LysM$^{Cre}$, $P < 0.0001$). (K–M) qRT-PCR analysis of the relative mRNA levels of *Fabp4* ((K), $n = 7–8$), *Pparγ* ((L), $n = 7–8$) and *C/ebpα* ((M), $n = 7–8$) in differentiated 3T3-L1 cells cultured with conditioned medium of BMDMs from EP3$^{F/F}$ and EP3$^{F/F}$LysM$^{Cre}$ mice transfected with Dnmt1 and Dnmt3a siRNA. (K–M) Scram EP3$^{F/F}$ vs Scram EP3$^{F/F}$LysM$^{Cre}$, Scram EP3$^{F/F}$ vs Si- Dnmt (1,3a) EP3$^{F/F}$, Scram EP3$^{F/F}$LysM$^{Cre}$ vs Si- Dnmt (1,3a) EP3$^{F/F}$LysM$^{Cre}$ in (K) ($P = 0.0005$; $P = 0.0099$; $P < 0.0001$), (L) ($P = 0.0071$; $P = 0.0005$; $P < 0.0001$), (M) ($P < 0.0001$; $P = 0.0002$; $P < 0.0001$). (N) Western blot analysis of Fabp4, Pparγ, C/EBPα in differentiated 3T3-L1 cells cultured with conditioned medium of BMDMs from EP3$^{F/F}$ and EP3$^{F/F}$LysM$^{Cre}$ mice transfected with Dnmt1 and Dnmt3a siRNA. Data information: Data represent the mean ± SEM. Data are representative of two independent experiments with biological replicates (D, F, J–M). Statistics: Mann–Whitney *U* test (C), unpaired Student's *t* test (D, F), two-way ANOVA (J–M). (C, D, F, J–M) *P* values are indicated by asterisks, with *$P < 0.05$, **$P < 0.01$, ***$P < 0.001$. Source data are available online for this figure.

intraperitoneally with D-glucose (1 g/kg body weight). For insulin tolerance test (ITT) experiments, mice were injected intraperitoneally with insulin (0.75 IU/kg body weight; #S20133006, Novolin R, Novo Nordisk, Denmark) following a 6-h fast. Blood glucose levels were monitored using a glucometer at 0, 30, 60, 90, 120, 150, and 180 min after injection (Accu-Chek Go, Roche, Mannheim, Germany).

## Macrophage, adipocytes and peripheral blood mononuclear cells (PBMCs) isolation

Preparation of mouse adipose tissue suspensions and flow cytometric sorting of macrophages were performed as previously reported (Moon et al, 2021). Briefly, murine epididymal WAT was harvested, minced on ice, and digested in phosphate-buffered saline (PBS) containing collagenase II (1 mg/mL) at 37 °C with shaking at 160 rpm for 30 min. The cell suspension was filtered through a 70-μm cell strainer (#22-363-548, Fisher Scientific, Billings, MT, USA), and centrifuged at 500×*g* for 5 min to pellet the stromal-vascular fraction (SVF) cells. SVF cells were resuspended, incubated in red blood cell (RBC) lysis buffer for 5 min, and washed twice with PBS. The SVF cells were resuspended in FACS staining buffer and stained with fluorescent-conjugated antibodies for 30 min at 4 °C in the dark. The antibodies were used as follows: anti-CD45-APCCy7 (#47-0451-82, 1:200; ebioscience, San Diego, CA, USA), anti-F4/80-Brilliant Violet 421 (#123137, 1:200; BioLegend, San Diego, CA, USA), anti-CD11b-FITC (#101206, 1:200; BioLegend). After staining, the cells were gently washed twice in ice-cold PBS and suspended in flow cytometry buffer (PBS containing 0.5% BSA and 2 mM EDTA). CD45$^+$F4/80$^+$CD11b$^+$ macrophages were sorted through a BD FACSAriaTM II flow cytometry system (BD Biosciences).

Isolation of Primary Murine Adipocytes were performed as previously reported (Curtin et al, 2025). In brief, murine epididymal WAT was Collected, minced, and digested in collagenase buffer at 37 °C with shaking at 220 rpm for 10 min. Add a volume of PBS equal to the volume of tissue and collagenase solution to neutralize collagenase. Spin the tubes at 300×*g* for 3 min, and the tissue separates into four layers, the topmost layer is the free lipids, the second layer is the mature adipocyte layer, the third layer is the collagenase and FBS, and the

fourth layer is the stromal vascular fraction (SVF) that pellets to the bottom. Mature adipocyte layer was collected to the prepared 15 mL tube with 10 mL of growth media after removing the top layer, and washed mature adipocyte layer two times. Finally, adipocytes were collected after removing free lipids by centrifuging cells at 100 × *g* for 1 min three times. PBMCs were isolated using the Mouse Peripheral Blood Mononuclear Cell Isolation Kit (#P5230, Solarbio, China) according to the manufacturer's protocol.

## Measurement of metabolic rate

As previously described (Hu et al, 2021), five-week-old male mice were fed an HFD for 3 weeks and placed in the metabolic chambers of the CLAMS (Columbus Instruments). The oxygen consumption rate (VO$_2$), carbon dioxide production (VCO$_2$) rate, energy expenditure, and food and water consumption were measured every 10 min over a 48-h period. Mice were housed in an environment with controlled temperature, a 12:12-h day–night cycle, and food and water were available.

## Hyperinsulinemic-euglycemic clamp

The experimental process was performed as described previously (Xu et al, 2016). Briefly, all mice were fasted for 4–5 h, mice were given an intravenous bolus injection and then continuously infused with human insulin at a rate of 60 pmol/kg/min for 2 h. Blood samples were obtained every 10 min for determination of glucose concentration. A 10% glucose solution was infused at variable rates that was adjusted to maintain the blood glucose concentration at the basal level (95 ± 5 mg/dl).

## Isolation, culturing, and transfection of BMDMs

Primary BMDMs were prepared as described previously (Yan et al, 2022) with slight modifications. Briefly, the tibia and femur were dissected from male mice aged 8–12 weeks, and the bone marrow was flushed out using a 1-mL syringe with Dulbecco's modified Eagle's medium (#C11885500; Gibco, Billings, MT, USA) supplemented with 1% penicillin/streptomycin (PS) (#V900929; Sigma-Aldrich) and 0.2% fetal bovine serum (FBS) (#10091155; Gibco).

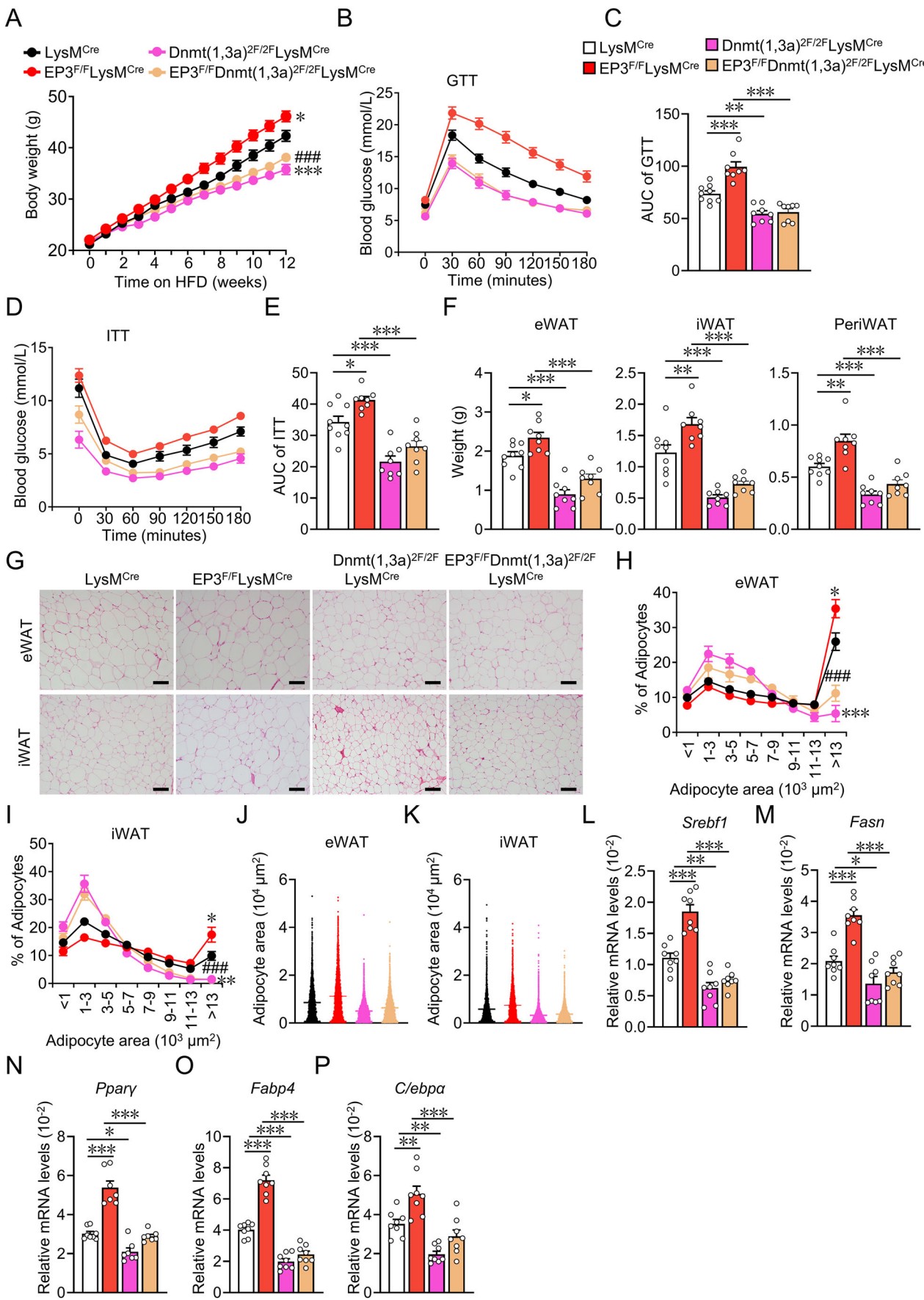

**Figure 6. Macrophage-specific Dnmt1/3a ablation blunted HFD-induced obesity in EP3$^{F/F}$LysM$^{Cre}$ mice.**

(A) Body weight analysis of LysM$^{Cre}$, EP3$^{F/F}$LysM$^{Cre}$, Dnmt(1,3a)$^{2F/2F}$LysM$^{Cre}$, and EP3$^{F/F}$Dnmt(1,3a)$^{2F/2F}$LysM$^{Cre}$ mice fed with HFD ($n = 8$–9) (12 weeks LysM$^{Cre}$ vs EP3$^{F/F}$LysM$^{Cre}$: $P = 0.0465$; 12 weeks LysM$^{Cre}$ vs Dnmt(1,3a)$^{2F/2F}$LysM$^{Cre}$: $P = 0.0003$; 12 weeks EP3$^{F/F}$LysM$^{Cre}$ vs EP3$^{F/F}$Dnmt(1,3a)$^{2F/2F}$LysM$^{Cre}$: $P < 0.0001$). (B) GTT in LysM$^{Cre}$, EP3$^{F/F}$LysM$^{Cre}$, Dnmt(1,3a)$^{2F/2F}$LysM$^{Cre}$, and EP3$^{F/F}$ Dnmt(1,3a)$^{2F/2F}$LysM$^{Cre}$ mice fed with HFD ($n = 8$–9). (C) AUC of the GTT measurement described in (B) ($n = 8$–9) (LysM$^{Cre}$ vs EP3$^{F/F}$LysM$^{Cre}$: $P < 0.0001$; LysM$^{Cre}$ vs Dnmt(1,3a)$^{2F/2F}$LysM$^{Cre}$: $P = 0.0021$; EP3$^{F/F}$LysM$^{Cre}$ vs EP3$^{F/F}$Dnmt(1,3a)$^{2F/2F}$LysM$^{Cre}$: $P < 0.0001$). (D) ITT in LysM$^{Cre}$, EP3$^{F/F}$LysM$^{Cre}$, Dnmt(1,3a)$^{2F/2F}$LysM$^{Cre}$, and EP3$^{F/F}$Dnmt(1,3a)$^{2F/2F}$LysM$^{Cre}$ mice fed with HFD ($n = 8$–9). (E) AUC of ITT described in (D) ($n = 8$–9) (LysM$^{Cre}$ vs EP3$^{F/F}$LysM$^{Cre}$: $P = 0.0282$; LysM$^{Cre}$ vs Dnmt(1,3a)$^{2F/2F}$LysM$^{Cre}$: $P < 0.0001$; EP3$^{F/F}$LysM$^{Cre}$ vs EP3$^{F/F}$Dnmt(1,3a)$^{2F/2F}$LysM$^{Cre}$: $P < 0.0001$). (F) The weights of adipose tissues of eWAT, iWAT and periWAT in LysM$^{Cre}$, EP3$^{F/F}$LysM$^{Cre}$, Dnmt(1,3a)$^{2F/2F}$LysM$^{Cre}$, and EP3$^{F/F}$Dnmt(1,3a)$^{2F/2F}$LysM$^{Cre}$ mice fed with HFD ($n = 8$–9). (F) LysM$^{Cre}$ vs EP3$^{F/F}$LysM$^{Cre}$, LysM$^{Cre}$ vs Dnmt(1,3a)$^{2F/2F}$LysM$^{Cre}$, EP3$^{F/F}$LysM$^{Cre}$ vs EP3$^{F/F}$Dnmt(1,3a)$^{2F/2F}$LysM$^{Cre}$ in eWAT graph ($P = 0.0443$; $P < 0.0001$; $P < 0.0001$), iWAT graph ($P = 0.0052$; $P < 0.0001$; $P < 0.0001$), PeriWAT graph ($P = 0.0019$; $P = 0.0007$; $P < 0.0001$). (G) Representative image of H&E staining for eWAT and iWAT from LysM$^{Cre}$, EP3$^{F/F}$LysM$^{Cre}$, Dnmt(1,3a)$^{2F/2F}$LysM$^{Cre}$, and EP3$^{F/F}$Dnmt(1,3a)$^{2F/2F}$LysM$^{Cre}$ mice fed with HFD (scale bar: 100 µm). (H–K) Quantification of adipocyte area of eWAT (H, J) and iWAT (I, K) in (G) ($n > 4565$ adipocytes measured from 6 to 8 mice in each group). (H, I) $> 13 \times 10^3$ µm$^2$ adipocyte LysM$^{Cre}$ vs EP3$^{F/F}$LysM$^{Cre}$, $> 13 \times 10^3$ µm$^2$ adipocyte LysM$^{Cre}$ vs Dnmt(1,3a)$^{2F/2F}$LysM$^{Cre}$, $>13 \times 10^3$ µm$^2$ adipocyte EP3$^{F/F}$LysM$^{Cre}$ vs EP3$^{F/F}$Dnmt(1,3a)$^{2F/2F}$LysM$^{Cre}$ in (H) ($P = 0.0469$; $P < 0.0001$; $P < 0.0001$), (I) ($P = 0.0142$; $P = 0.0063$; $P < 0.0001$). (L–P) qRT-PCR analysis of the relative mRNA levels of fatty acid synthesis and adipogenesis genes in eWAT from LysM$^{Cre}$, EP3$^{F/F}$LysM$^{Cre}$, Dnmt(1,3a)$^{2F/2F}$LysM$^{Cre}$, and EP3$^{F/F}$Dnmt(1,3a)$^{2F/2F}$LysM$^{Cre}$ mice ($n = 7$–8). (L–P) LysM$^{Cre}$ vs EP3$^{F/F}$LysM$^{Cre}$, LysM$^{Cre}$ vs Dnmt(1,3a)$^{2F/2F}$LysM$^{Cre}$, EP3$^{F/F}$LysM$^{Cre}$ vs EP3$^{F/F}$Dnmt(1,3a)$^{2F/2F}$LysM$^{Cre}$ in (L) ($P < 0.0001$; $P = 0.0014$; $P < 0.0001$), (M) ($P < 0.0001$; $P = 0.042$; $P < 0.0001$), (N) ($P < 0.0001$; $P = 0.0153$; $P < 0.0001$), (O) ($P < 0.0001$; $P < 0.0001$; $P < 0.0001$), (P) ($P = 0.0046$; $P = 0.0039$; $P < 0.0001$). Data information: Data represent the mean ± SEM. Data are pooled from two independent experiments with biological replicates (A–F, H–P). Statistics: two-way ANOVA (A, C, E, F, H, I, L–P). (A, C, E, F, H, I, L–P) $P$ values are indicated by asterisks, with *$P < 0.05$, **$P < 0.01$, ***$P < 0.001$ vs. LysM$^{Cre}$, ###$P < 0.001$ vs. EP3$^{F/F}$LysM$^{Cre}$. Source data are available online for this figure.

The cell suspension was centrifuged at $300 \times g$ for 5 min, and the pellets were resuspended with RBC lysis buffer and incubated at room temperature for 3–5 min. Subsequently, the cell suspension was filtered through a 40-µm sterile cell strainers (#22-363-547; Fisher Scientific) and centrifuged at $300 \times g$ for 5 min. Bone marrow cells were resuspended in BMDMs condition culture medium (DMEM supplemented with 10% FBS, 1% PS, and 20% L929-cell (murine M-CSF-producing cell line)-conditioned medium and cultured at 37 °C with 5% CO$_2$ for 7 days. The BMDM-conditioned culture medium was changed on day 3 of differentiation. For metabolically-activated macrophage (MMe activation), BMDMs were stimulated with a combination of glucose (30 mM), insulin (10 nM), and palmitate (0.4 mM) for different periods (Kratz et al, 2014).

BMDMs were transfected with 50 pmol of each siRNA with the use of Lipofectamine™ RNAiMAX Transfection Reagent (#13778075; Invitrogen, USA) in 1 ml of Opti-MEM serum-reduced medium in accordance with the manufacturer's instructions. The cells were collected for further analysis after 48–72 h of transfection. All siRNA sequences (Hanbio Biotechnology, Shanghai, China) are listed in Appendix Table S1.

## PG extraction and analysis

Mouse eWAT (approximately 40 mg) sample was homogenized with mixture solution at 4 °C consisting of methanol (500 µL), acetic acid (10 µL), 0.01 M butylated hydroxytoluene (10 µL) and an internal standard (1 µL). After centrifugation at $12,000 \times g$ for 10 min, the supernatant was collected. Subsequently, 700 µL water and 1 mL methyl tert-butyl ether (MTBE) were added to the supernatant, strenuously vibrated for 2 min, and allowed to stand for 10 min. The supernatant of the organic phase was collected after centrifugation at $4000 \times g$ for 10 min. The organic layer was evaporated under nitrogen flow, redissolved in 100 µL water with 30% acetonitrile, vortex-mixed for 5–10 min, and passed through small centrifugal filters with the 0.2-µm nylon membrane. Prostanoid metabolites were quantified using liquid chromatography-tandem mass spectrometry (LC-MS/MS). PG production was normalized to the tissue weight.

## RNA extraction and real-time PCR

Total RNA was extracted from tissues and cells using TRIzol reagent (#15596018; Invitrogen, Carlsbad, CA, USA) and reverse-transcribed into cDNA using Hifair® III 1st Strand cDNA Synthesis SuperMix for qPCR (gDNA digester plus) (#11141ES60; YeaSen Biotechnology, Shanghai, China) according to the manufacturer's instruction. Real-time PCR analyses were performed with Hieff UNICON® Universal Blue qPCR SYBR Green Master Mix (#11184ES03; YeaSen Biotechnology) using a Roche LightCycler 480 Instrument II. GAPDH was used as an internal control to normalize relative gene expression. The primer sequences are listed in Appendix Table S2.

## Western blotting

Protein lysates from isolated tissues or cultured cells were extracted using a lysis buffer containing protease and phosphatase inhibitors. The protein concentration was measured using a bicinchoninic acid Protein Assay Kit (#23227; Thermo Scientific, Waltham, MA, USA). Equal amounts of total protein were separated using SDS-PAGE and transferred to a polyvinylidene difluoride membrane (#IPVH00010; Millipore, Billerica, MA, USA). The membranes were incubated for 2 h in blocking solution (5% non-fat milk powder in tris-buffered saline-tween 20 (TBST)), then incubated with primary antibody (in blocking solution) overnight at 4 °C. The primary antibodies were diluted as follows: rabbit anti-Dnmt1 (#5032, 1:1000; Cell Signaling Technology, Boston, MA, USA), rabbit anti-Dnmt3a (#ab188470, 1:2000; Abcam, Cambridge, Cambs, UK), rabbit anti-Dnmt3b (#ab2851, 1:500; Abcam), rabbit anti-SPARC (#5420, 1:1000; Cell Signaling Technology), rabbit anti-Sp1 (#21962-1-AP, 1:2000; Proteintech, Rosemont, IL, USA), rabbit anti-phospho-Sp1 (T453) (#ab59257, 1:500; Abcam), rabbit anti-TET1 (#A1506, 1:1000; abclonal), rabbit anti-TET2 (#21207-1-AP, 1:1000; Proteintech), rabbit anti-TET3 (#A7612, 1:1000; abclonal), rabbit anti-NF-κb p65 (#8242S, 1:1000; Cell Signaling Technology), rabbit Histone H3 (#4499S, 1:2000; Cell Signaling Technology), rabbit anti-NOS2 (#ab15323, 1:200; Abcam), goat anti-YM1 (#AF2446, 1:1000; R&D Systems), mouse anti-α-tubulin

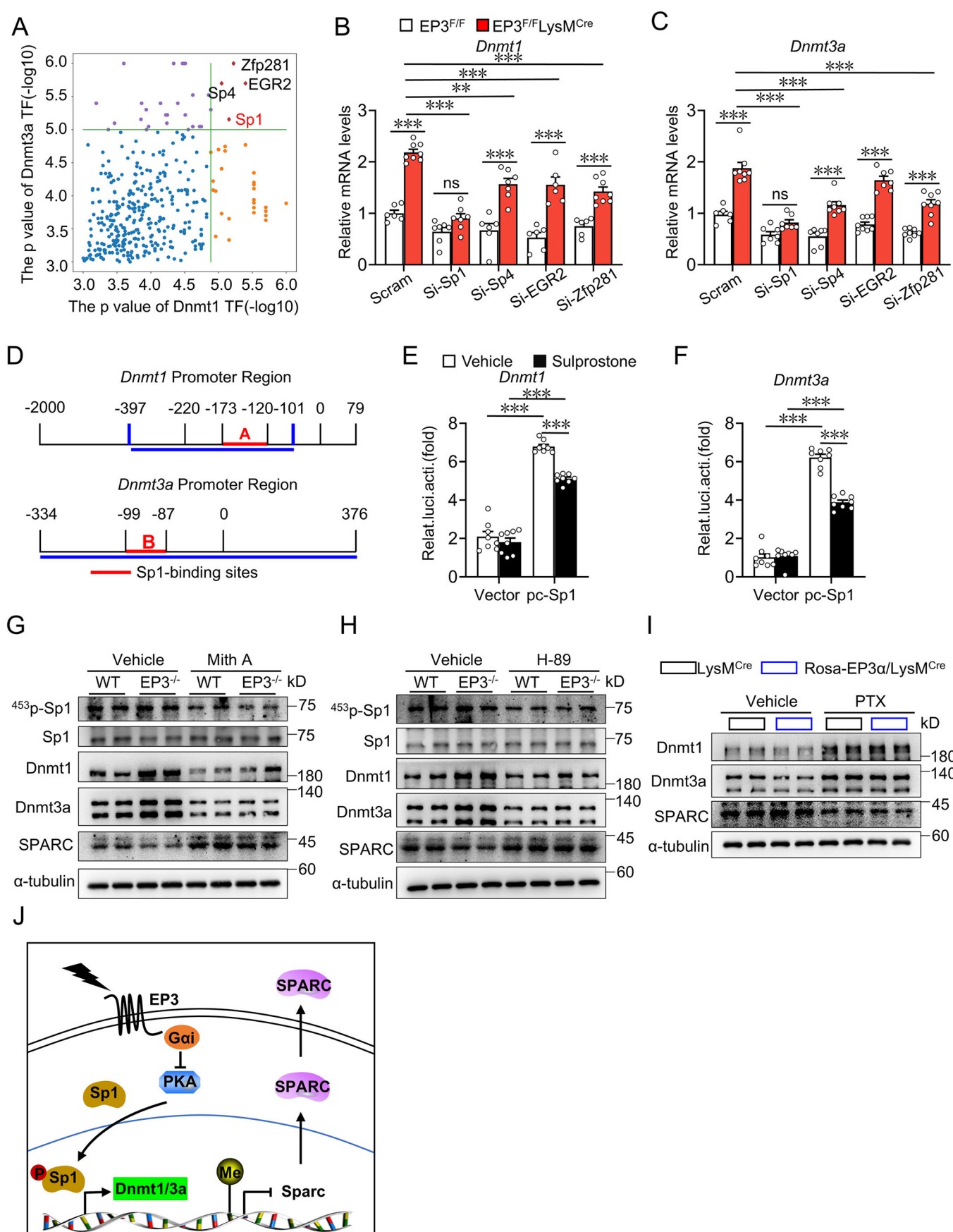

Figure 7. EP3 deficiency promoted Dnmt1/3a expression via PKA-mediated Sp1 phosphorylation.

(A) The P values of transcription factors controlling Dnmt1 and Dnmt3a expression were predicted by TF2DNA Database. (B, C) qRT-PCR analysis of the relative mRNA levels of *Dnmt1* (B) (n = 6–8) and *Dnmt3a* (C) (n = 6–8) in BMDMs from EP3$^{F/F}$ and EP3$^{F/F}$LysM$^{Cre}$ mice transfected with Sp1, Sp4, EGR2, and Zfp281 siRNA. (B) Scram EP3$^{F/F}$ vs Scram EP3$^{F/F}$LysM$^{Cre}$, P < 0.0001; Scram EP3$^{F/F}$LysM$^{Cre}$ vs Si-Sp1 EP3$^{F/F}$LysM$^{Cre}$, P < 0.0001; Si-Sp1 EP3$^{F/F}$ vs Si-Sp1 EP3$^{F/F}$LysM$^{Cre}$, P = 0.095; Scram EP3$^{F/F}$LysM$^{Cre}$ vs Si-Sp4 EP3$^{F/F}$LysM$^{Cre}$, P = 0.0011; Si-Sp4 EP3$^{F/F}$ vs Si-Sp4 EP3$^{F/F}$LysM$^{Cre}$, P < 0.0001; Scram EP3$^{F/F}$LysM$^{Cre}$ vs Si-EGR2 EP3$^{F/F}$LysM$^{Cre}$, P = 0.0005; Si-EGR2 EP3$^{F/F}$ vs Si-EGR2 EP3$^{F/F}$LysM$^{Cre}$, P < 0.0001; Scram EP3$^{F/F}$LysM$^{Cre}$ vs Si-Zfp281 EP3$^{F/F}$LysM$^{Cre}$, P < 0.0001; Si-Zfp281 EP3$^{F/F}$ vs Si-Zfp281 EP3$^{F/F}$LysM$^{Cre}$, P < 0.0001. (C) Scram EP3$^{F/F}$ vs Scram EP3$^{F/F}$LysM$^{Cre}$, P < 0.0001; Scram EP3$^{F/F}$LysM$^{Cre}$ vs Si-Sp1 EP3$^{F/F}$LysM$^{Cre}$, P < 0.0001; Si-Sp1 EP3$^{F/F}$ vs Si-Sp1 EP3$^{F/F}$LysM$^{Cre}$, P = 0.1871; Scram EP3$^{F/F}$LysM$^{Cre}$ vs Si-Sp4 EP3$^{F/F}$LysM$^{Cre}$, P < 0.0001; Si-Sp4 EP3$^{F/F}$ vs Si-Sp4 EP3$^{F/F}$LysM$^{Cre}$, P = 0.0001; Si-EGR2 EP3$^{F/F}$ vs Si-EGR2 EP3$^{F/F}$LysM$^{Cre}$, P < 0.0001; Scram EP3$^{F/F}$LysM$^{Cre}$ vs Si-Zfp281 EP3$^{F/F}$LysM$^{Cre}$, P < 0.0001; Si-Zfp281 EP3$^{F/F}$ vs Si-Zfp281 EP3$^{F/F}$LysM$^{Cre}$, P = 0.0001. (D) Schematic illustration of luciferase reporter containing truncated mouse Dnmt1 and Dnmt3a promoter region. (E, F) Effect of EP3 agonist on mouse Dnmt1 (E) (n = 8) and Dnmt3a (F) (n = 8) promoter fragment-mediated luciferase activity in Sp1 overexpressed 293T cells. (E) Every comparison P < 0.0001. (F) Every comparison P < 0.0001. (G, H) Western blot analysis of Sp1 phosphorylation, Sp1, Dnmt1, Dnmt3a, SPARC in WT and EP3$^{-/-}$ BMDMs with treatment of Sp1 inhibitor (G) and PKA inhibitor (H). (I) Western blot analysis of Dnmt1, Dnmt3a, SPARC in BMDMs from LysM$^{Cre}$ and Rosa-EP3α/LysM$^{Cre}$ mice with treatment of G$_{αi}$ inhibitor. (J) Schematic illustration of EP3/Sp1/Dnmt1/3a/ SPARC pathway-mediated adipocyte differentiation in macrophage. Data information: Data represent the mean ± SEM. Data are representative of two independent experiments with biological replicates (B, C, E, F). Statistics: two-way ANOVA (B, C, E, F). (B, C, E, F) P values are indicated by asterisks, with **P < 0.01, ***P < 0.001. Source data are available online for this figure.

(#KM9007,1:2000; Sungene Biotech). After washing with TBST thrice for 10 min, the membranes were incubated with horseradish peroxidase (HRP)-conjugated secondary antibodies (#7074S, #7076S, 1:2000; Cell Signaling Technology) and HRP-conjugated Rabbit anti-Goat IgG (#AS029, 1:5000; abclonal) in blocking buffer for 2 h at room temperature. After washing the membrane with TBST, the blots were developed using an enhanced chemiluminescence reagent (#34580; Thermo Scientific) and scanned using a Tanon Imaging System (Tanon5200Multi, Shanghai, China).

## Coculture of adipocytes with BMDMs

The coculture of 3T3-L1 preadipocytes and BMDMs in an indirect coculture transwell system (#3450, CORNING, Corning, NY, USA) was performed as described previously (Suganami et al, 2005) with slight modifications. Briefly, BMDMs were cultured and treated in the upper chamber containing the 0.4 µm porous membrane. 3T3-L1 preadipocytes were plated in the lower chamber of the six-well plate and differentiated with a hormonal cocktail (1 µM Dexamethasone, 0.5 mM isobutyl-l-methylxanthine, and 5 µg/mL insulin) in medium for 2 days. Then the medium was replaced with DMEM containing 5 µg/mL insulin, and changed every 2 to 3 days until cells were collected for analysis. Lipid accumulation in differentiated cells was visualized using Oil Red O staining (#G1015, Servicebio, Wuhan, Hubei, China) according to the manufacturer's instructions, and quantified using ImageJ software.

## DNA methylation analysis of SPARC

DNA was extracted from mouse BMDMs using a DNA extraction kit (#DP304-03; TIANGEN, Beijing, China). Genomic DNA was then subjected to bisulfite conversion using an EpiTect Fast DNA Bisulfite Kit (#59826; QIAGEN, Venlo, Netherlands) following the manufacturer's protocol. The specific bisulfite-converted region was amplified by PCR. Bisulfite-specific primers were designed, and their sequences are provided in Appendix Table S3. The resulting PCR products were purified, cloned into the pGEM-T Easy vector, and transformed into XL10-Gold. Positive clones were selected and sequenced by GENERAL BIOL, Inc.

## Histology and immunofluorescence staining

The adipose tissues were dissected and fixed in a special fat-fixing solution (#G1119, Servicebio) for at least 24 h at room temperature. The fixed tissues were dehydrated through a graded series of ethanol and embedded in paraffin, sectioned into 5-µm-thick slices, and stained with hematoxylin and eosin (H&E) and Masson's trichrome followed by microscopy. Images were captured using a Leica Olympus BX51 microscope (Leica Microsystems, Wetzlar, Germany) and the adipocyte area was quantified using ImageJ software. For immunofluorescence, sections were deparaffinized and rehydrated, and antigen retrieval was performed by boiling in citrate buffer for 20 min. The sections were cooled to room temperature, blocked with goat serum for 1 h, and incubated with primary antibodies against Mac-3 (#108512, 1:200; Biolegend); Dnmt1 (#ab188453, 1:100; Abcam); Dnmt3a (#188470, 1:1000; Abcam); SPARC (#8725S, 1:1000; Cell Signaling Technology); CD31(#ab28364, 1:50; Abcam) at 4 °C overnight. An isotype IgG antibody was used as a negative control to validate antibody specificity. The slides were washed with PBS and incubated with Alexa Fluor 488 conjugated secondary antibodies (#A11008, 1:500; Invitrogen) or Alexa Fluor 647 conjugated secondary antibodies (#112-606-07,1:500; Jackson ImmunoResearch, West Grove, PA, USA) for 2 h at room temperature. Nuclei were counterstained with DAPI for 5 min. Prolonged gold antifade reagent with DAPI (Invitrogen) was used to mount and counterstain the slides. All images were captured using a laser confocal ultra-high-resolution microscope (LSM900) and analyzed using ImageJ. All representative images and figures were chosen according to the mean values of the results for each group.

## Enzyme-linked immunosorbent assay (ELISA)

Levels of cytokines in adipose tissue were measured using mouse ELISA for IL-6 (#E-EL-M0044, Elabscience), TNF-α (#E-EL-M3063, Elabscience), and MCP-1 (#E-EL-M3001, Elabscience). ELISAs were performed according to the manufacturer's instructions.

## Luciferase reporter assay

Empty vector or Dnmt1 and Dnmt3a luciferase reporters, together with Renilla luciferase plasmid-thymidine kinase (pRL-TK), were transfected into 293T cells using the Lipofectamine 2000 transfection reagent (#11668019, Invitrogen) according to the manufacturer's instructions. The cells were then lysed and luciferase activity was monitored using the Dual-Luciferase Reporter Assay System (#E1960, Promega, Madison, WI, USA) according to the manufacturer's instructions. The DNA fragments of Dnmt1 promoter

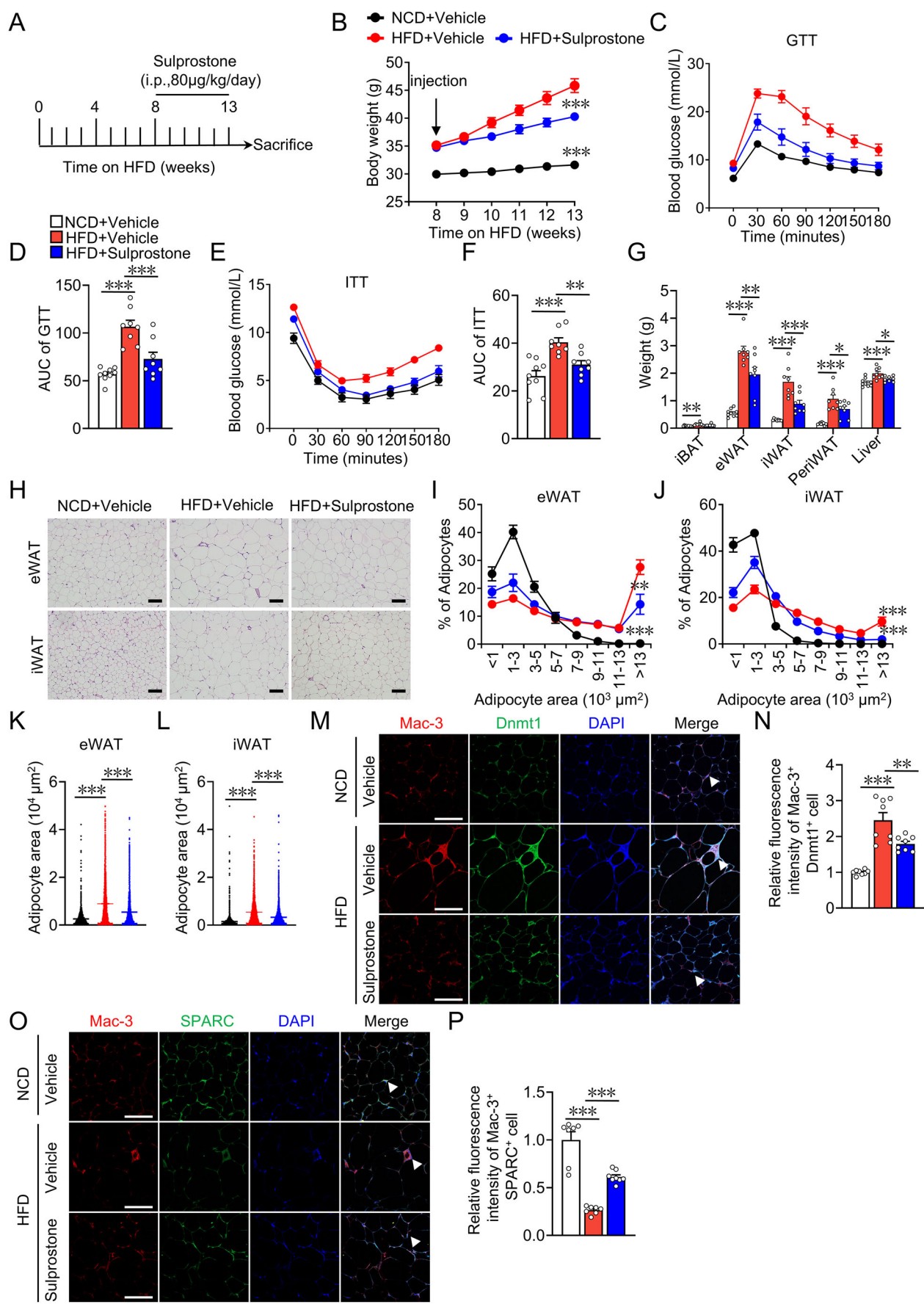

Figure 8.   EP3 agonist sulprostone treatment attenuated HFD-induced obesity in mice.

(A) Protocol for administration of EP3 agonist sulprostone to HFD-challenged mice. (B) Body weight analysis of HFD-challenged mice with or without sulprostone treatment ($n = 8$–10) (13 weeks HFD+Vehicle vs NCD+Vehicle, $P < 0.0001$; 13 weeks HFD+Vehicle vs HFD+Sulprostone, $P = 0.0003$). (C) GTT in HFD-challenged mice with or without sulprostone treatment ($n = 8$–9). (D) AUC of GTT described in (C) ($n = 8$–9) (HFD+Vehicle vs NCD+Vehicle, $P < 0.0001$; HFD+Vehicle vs HFD+Sulprostone, $P = 0.0007$). (E) ITT in HFD-challenged mice with or without sulprostone treatment ($n = 8$–9). (F) AUC of ITT described in (E) ($n = 8$–9) (HFD+Vehicle vs NCD+Vehicle, $P < 0.0001$; HFD+Vehicle vs HFD+Sulprostone, $P = 0.006$). (G) The weights of adipose tissues of iBAT, eWAT, iWAT, periWAT and liver in HFD-challenged mice with or without sulprostone treatment ($n = 8$–9). (G) HFD+Vehicle vs NCD+Vehicle, HFD+Vehicle vs HFD+Sulprostone in iBAT graph ($P = 0.0059$; $P = 0.1$), eWAT graph ($P < 0.0001$; $P = 0.0059$), iWAT graph ($P < 0.0001$; $P = 0.0006$), PeriWAT ($P < 0.0001$; $P = 0.0326$), Liver graph ($P = 0.0006$; $P = 0.0332$). (H) Representative image of H&E staining for eWAT and iWAT from normal chow diet and HFD-challenged mice with or without sulprostone treatment (scale bar: 100 µm). (I–L) Quantification of adipocyte area of eWAT (I, K) and iWAT (J, L) in (H) ($n > 5848$ adipocytes measured from 7–9 mice in each group). (I, J) >$13 \times 10^3$ µm$^2$ adipocyte HFD+Vehicle vs NCD+Vehicle, >$13 \times 10^3$ µm$^2$ adipocyte HFD+Vehicle vs >HFD+Sulprostone in (I) ($P < 0.0001$; $P = 0.0022$), (J) ($P < 0.0001$; $P = 0.0005$). (K) Every comparison $P < 0.0001$. (L) Every comparison $P < 0.0001$. (M) Representative images of Mac-3, Dnmt1 immunostaining of the eWAT from HFD-challenged mice with or without sulprostone treatment (scale bar: 50 µm). White arrowheads indicate Mac-3$^+$Dnmt1$^+$cells. (N) Quantification of Dnmt1 fluorescence intensity in (M) ($n = 8$) (HFD+Vehicle vs NCD+Vehicle, $P < 0.0001$; HFD+Vehicle vs HFD+Sulprostone, $P = 0.0034$). (O) Representative images of Mac-3, SPARC immunostaining of the eWAT from HFD-challenged mice with or without sulprostone treatment (scale bar: 50 µm). White arrowheads indicate Mac-3$^+$SPARC$^+$ cells. (P) Quantification of SPARC fluorescence intensity in (O) ($n = 7$–8) (HFD+Vehicle vs NCD+Vehicle, $P < 0.0001$; HFD+Vehicle vs HFD+Sulprostone, $P = 0.0002$). Mac-3 antibody was utilized to identify and visualize ATMs. Data information: Data represent the mean ± SEM. Data are pooled from two independent experiments with biological replicates (B–G, I–L, N, P). Statistics: one-way ANOVA (B, D, F, G, I–L, N, P). (B, D, F, G, I–L, N, P) $P$ values are indicated by asterisks, with *$P < 0.05$, **$P < 0.01$, ***$P < 0.001$. Source data are available online for this figure.

(−397 to −101 bp) and Dnmt3a promoter (−334 to 376 bp) were synthetized and separately subcloned into pGL3-basic vector (Hanheng Biotechnology, Shanghai, China).

## 4D-label-free proteomics analyses

4D-label-free proteomics analyses were conducted as described (Fang et al, 2024). Culture medium supernatants from palmitate-treated WT and EP3$^{-/-}$ BMDMs were collected and concentrated using centrifugal filters (#UFC901096; Millipore Sigma). SDT buffer (4% SDS, 100 mM Tris-HCl, pH 7.6) was added to each sample for protein extraction, and trypsin was used to digest the protein according to the filter-aided sample preparation procedure proposed by Matthias Mann (Wisniewski et al, 2009). The digest peptides of each sample were desalted, concentrated, and reconstituted in 40 µL 0.1% (v/v) formic acid. LC-MS/MS analysis was performed using a timeTOF Pro mass spectrometer (Bruker, Billerica, MA, USA) coupled to a nanoelute (Bruker). Identification and quantitative analysis of protein expression were performed using MaxQuant 1.5.3.17 from the mass spectrum data.

The prediction of secreted proteins and bioinformatics analysis were performed as described previously (Fu et al, 2018). Signal P 4.1 was first classified into classical and nonclassical signals from the secretory proteins in all quantified proteins. TMHMM server 2.0 was analyzed with classical signal secretory proteins (TMHs = 0 was considered as a classical secretory protein and TMHs ≥1 was standard for a membrane protein). Secretome P 2.0 analysis was calculated a neural network score (NN-score) of nonclassical signal proteins (an NN score ≥0.5 was represented as secretory proteins). The Mus Exocarta was analyzed to avoid incorrect filtering of proteins secreted without a classical or nonclassical signal by transportation of exosome vesicles. Proteins with a fold change >2 and $P$ value < 0.05 between the two groups were recognized as differentially expressed proteins. Gene ontology (GO) enrichment analyses were performed using DAVID.

## Single-cell RNA sequencing data analysis

The data for this study were extracted from the GEO database under the accession numbers (GSE176067, GSE176171) (Emont

et al, 2022). Specifically, we focused on human visceral adipose tissue (VAT) macrophages. The dataset contains single-cell RNA-sequencing data from multiple individuals with varying BMI measurements. Prior to downstream analyses, the raw gene expression counts were subjected to log transformation to stabilize the variance. The log-transformed counts were then used for all subsequent analyses (Ahlmann-Eltze and Huber, 2023). For the purpose of this study, we categorized the human VAT macrophages into two groups based on BMI: Cells from individuals with a BMI between 20–30 and 30–40. To identify differentially expressed genes between two groups, we employed the Mann–Whitney $U$ test.

## Statistical analyses

All data were presented as the mean ± standard error of the mean (SEM). Statistical analyses were performed using GraphPad Prism 8 software (GraphPad Prism Software Inc., San Diego, CA, USA). For datasets with sample sizes ≥6, normality was assessed using the Shapiro–Wilk test. For normally distributed data, an unpaired two-tailed Student's $t$ test was applied. For non-normally distributed data or when sample sizes were <6, the Mann–Whitney $U$ test was utilized. Multiple comparisons were assessed using one- or two-way analysis of variance (ANOVA) by Dunnett's or Tukey's test for post hoc comparisons, as appropriate. Statistical significance was set at $P < 0.05$ for all tests. The researchers were blinded to the genotype and treatment of the mice during the experiment and data evaluation.

# Data availability

The mass spectrometry proteomics data have been deposited to ProteomeXchange via the PRIDE database with the accession number: PXD063443.

The source data of this paper are collected in the following database record: biostudies:S-SCDT-10_1038-S44318-025-00508-y.

# Peer review information

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

## Acknowledgements

This study was supported by the National Natural Science Foundation of China (82321001, 82030015, 82241016, 82261160656, 82300473, 82470447). Ying Yu is a fellow at the Jiangsu Collaborative Innovation Center for Cardiovascular Disease Translational Medicine in Jiangsu, China. Some cartoon components in this article were from www.figdraw.com for model drawing.

## Author contributions

**Wenlong Shang**: conceived, designed the research, performed the experiments, analyzed the data. **Yinxiu Li**: performed the experiments, analyzed the data. **Lu Wang**: biological advice. **Jiao Liu**: provided experimental assistance. **Huiwen Ren**: provided experimental assistance. **Qian Liu**: provided experimental assistance. **Shumin Guo**: provided experimental assistance. **Yuhong Wang**: provided experimental assistance. **Yubo Ma**: provided experimental assistance. **Tianyi You**: biological advice. **Yujun Shen**: provided help with the conceptual framework and manuscript writing for the study. **Yu Zhou**: provided the experimental resources. **Danyang Tian**: revised the manuscript for key intellectual content. **Ying Yu**: Writing—review and editing; conceived and designed the research, revised the manuscript for key intellectual content.

Source data underlying figure panels in this paper may have individual authorship assigned. Where available, figure panel/source data authorship is listed in the following database record: biostudies:S-SCDT-10_1038-S44318-025-00508-y.

## Disclosure and competing interests statement

The authors declare no competing interests.

