## [Peer Review File · The EMBO Journal]

Deletion of EP3 prostaglandin receptor in murine macrophages aggravates diet-induced obesity by suppressing SPARC

Wenlong Shang, Yinxiu Li, Lu Wang, Jiao Liu, huiwen ren, Qian Liu, Shumin Guo, Yuhong Wang, Yubo Ma, Tianyi You, Yujun Shen, Yu Zhou, Danyang Tian, and Ying Yu

Corresponding authors: Ying Yu (yuying@tmu.edu.cn) , Danyang Tian (tiandanyang@hebmu.edu.cn)

Review Timeline:

Submission Date:	12th Nov 24
Editorial Decision:	17th Dec 24
Revision Received:	6th May 25
Editorial Decision:	20th May 25
Revision Received:	24th May 25
Accepted:	2nd Jul 25

Editor: Daniel Klimmeck

Transaction Report:

Dear Dr Yu,

Thank you again for the submission of your manuscript (EMBOJ-2024-119612) to The EMBO Journal. As mentioned earlier, your study was assessed by three reviewers with expertise in control of body metabolism and obesity, immunity as well as adipocyte biology, whose comments are enclosed below.

As you will see from the experts' reports, the referees acknowledge the analysis and potential interest and value of your findings. However, they also express important issues regarding the completeness of your study of the results, which need to be addressed thoroughly to make them supportive of publication in the EMBO Journal. Further, the reviewers raise a number of issues related to the presentation of the findings, additional controls and improved methods annotation required, statistics applied and overall discussion of related literature, that would need to be conclusively addressed to achieve the level of robustness and clarity needed for The EMBO Journal.

Given the overall interest stated and broader angle of your findings, we are able to invite you to revise your manuscript experimentally to address the referees' comments. I need to stress though that we do require strong support from the referees on a revised version of the study in order to move on to publication of the work.

I would appreciate if you could contact me during the next weeks for exchange e.g. a video call to discuss your perspective on the comments and potential plan for revisions.

Please feel free to contact me if you have any questions or need further input on the referee comments.

When submitting your revised manuscript, please carefully review the instructions below.

Please feel free to approach me any time should you have additional questions related to this.

Thank you for the opportunity to consider your work for publication.

I look forward to your revision.

Kind regards,

Daniel Klimmeck

Daniel Klimmeck, PhD
Senior Editor
The EMBO Journal

Instruction for the preparation of your revised manuscript:

- 1) a .docx formatted version of the manuscript text (including legends for main figures, EV figures and tables). Please make sure that the changes are highlighted to be clearly visible.
- 2) individual production quality figure files as .eps, .tif, .jpg (one file per figure).
- 3) a .docx formatted letter INCLUDING the reviewers' reports and your detailed point-by-point response to their comments. As part of the EMBO Press transparent editorial process, the point-by-point response is part of the Review Process File (RPF), which will be published alongside your paper.
- 4) a complete author checklist, which you can download from our author guidelines ([https://wol-prod-cdn.literatumonline.com/pb-assets/embo-site/Author Checklist%20-%20EMBO%20J-1561436015657.xlsx](https://wol-prod-cdn.literatumonline.com/pb-assets/embo-site/Author%20Checklist%20-%20EMBO%20J-1561436015657.xlsx)). Please insert information in the checklist that is also reflected in the manuscript. The completed author checklist will also be part of the RPF.

6) It is mandatory to include a 'Data Availability' section after the Materials and Methods. Before submitting your revision, primary datasets produced in this study need to be deposited in an appropriate public database, and the accession numbers and database listed under 'Data Availability'. Please remember to provide a reviewer password if the datasets are not yet public (see <https://www.embopress.org/page/journal/14602075/authorguide#datadeposition>).

7) Our journal encourages inclusion of *data citations in the reference list* to directly cite datasets that were re-used and obtained from public databases. Data citations in the article text are distinct from normal bibliographical citations and should directly link to the database records from which the data can be accessed. In the main text, data citations are formatted as follows: "Data ref: Smith et al, 2001" or "Data ref: NCBI Sequence Read Archive PRJNA342805, 2017". In the Reference list, data citations must be labeled with "[DATASET]". A data reference must provide the database name, accession number/identifiers and a resolvable link to the landing page from which the data can be accessed at the end of the reference. Further instructions are available at .

8) At EMBO Press we ask authors to provide source data for the main and EV figures. Our source data coordinator will contact you to discuss which figure panels we would need source data for and will also provide you with helpful tips on how to upload and organize the files.

Numerical data can be provided as individual .xls or .csv files (including a tab describing the data). For 'blots' or microscopy, uncropped images should be submitted (using a zip archive or a single pdf per main figure if multiple images need to be supplied for one panel). Additional information on source data and instruction on how to label the files are available at .

9) We replaced Supplementary Information with Expanded View (EV) Figures and Tables that are collapsible/expandable online (see examples in <https://www.embopress.org/doi/10.15252/embj.201695874>). A maximum of 5 EV Figures can be typeset. EV Figures should be cited as 'Figure EV1, Figure EV2" etc. in the text and their respective legends should be included in the main text after the legends of regular figures.

11) For data quantification: please specify the name of the statistical test used to generate error bars and P values, the number (n) of independent experiments (specify technical or biological replicates) underlying each data point and the test used to calculate p-values in each figure legend. The figure legends should contain a basic description of n, P and the test applied. Graphs must include a description of the bars and the error bars (s.d., s.e.m.).

We realize that it is difficult to revise to a specific deadline. In the interest of protecting the conceptual advance provided by the work, we recommend a revision within 3 months (17th Mar 2025). Please discuss the revision progress ahead of this time with the editor if you require more time to complete the revisions.

Referee #1:

In this paper, authors tried to characterize the role of macrophage EP3 in adipose tissue. Analyzing macrophage-EP3 depletion model, authors concluded dietary obesity reduced EP3 in macrophage that led to a reduction in SPARC level in this type of cell which aggravated diet-induced obesity. Authors also showed EP3 reduced DNA methylation in macrophage that led to an increase in SPARC level and amelioration of HFD-induced obesity. Major issue relates with the lack of the characterization of inflammation. Recently, SPARC was shown to have pro-inflammatory role in adipose tissue (J Clin Invest. 2023 Oct 2;133(19):e169173. doi: 10.1172/JCI169173., Immunity. 2022. Sep 13;55(9):1609-1626.e7. doi: 10.1016/j.immuni.2022.07.007.). As authors may know, chronic sterile inflammation has critical roles for systemic glucose/ insulin intolerance and unhealthy obesity. These should be characterized in the white adipose tissues of models studied in this paper. Another potential issue is the contribution of macrophage-SPARC in the regulation of adipose SPARC level. The above J Clin Invest paper indicated the level of SPARC as rather low in monocytes compared to adipocytes. It is predicted that the level of SPARC in whole adipose tissue may be regulated mostly by adipocytes, and contribution of macrophage derived SPARC as minor. Authors should show the level of protein SPARC in the visceral fat of the 1) macrophage-EP3 overexpression model, 2) macrophage-EP3 knockout model.

Minor issues.

- 1) In Fig1B, what type of cells or tissues produces PGE2? What is the mechanism that increases this type of prostaglandin?
- 2) Related with Fig. 1C, what is the protein EP3 level in the macrophage of the visceral fat?
- 3) In Fig. 1D, how PA reduces EP3 level? The mechanisms need to be shown.
- 4) Fig. 2C and all other panels for GTT that are done under different body weight among genotypes should be done with clamp studies.
- 5) Again, here I would like to emphasize it as critically important to characterize the protein level of SPARC in the visceral fat of the 1) macrophage-EP3 overexpression model, 2) macrophage-EP3 knockout model.
- 6) In Fig. 3J, what is the protein SPARC level?
- 7) Fig. 5N indicates PPAR-gamma reduced in 3T3-L1 adipocytes administrated with conditioned medium collected from the si-Dnmt treated BMDMs. The following paper indicates inhibition of PPAR-gamma in adipocyte caused insulin resistance (Proc Natl Acad Sci U S A. 2003 Dec 23;100(26):15712-7. doi: 10.1073/pnas.2536828100. Epub 2003 Dec 5.), but Fig. 6A, B and D indicate the opposite phenotype. What is the role of adipocyte- PPAR-gamma in this paper? In the core model, PPAR-gamma also needs to be characterized in protein level in cell specific manner.
- 8) In all figures, figures should be alphabetically demonstrated (for example, not as Fig. 6E, F, H, G, J, K). Group of mice should be clearly labelled in all panels.
- 9) In the Fig. 8M, O, and EV7H, what do these arrows indicate?
- 10) In Fig. EV2B, why the RER increase in the KO mice only at the light time? What is the mechanisms?
- 11) What is the unit for EV3B?
- 12) What are the level of proinflammatory cytokines in the studied models? Together with transcriptome studies, some core molecules should be studied in protein levels.

Referee #2:

The manuscript by Shang et al. reported an important role of the PGE2-EP3 axis in macrophages in controlling adipogenesis and obesity pathogenesis. Combing in vitro and various in vivo loss- or gain-of-function mouse models, authors elegantly demonstrated that macrophage-derived PGE2 exerts an autocrine role through EP3 receptor to upregulate SPARC secretion, which exerts downstream anti-adipogenic effects. The underlying molecular mechanisms involved the PKA/SP1/DNMT signaling pathway. This is physiologically important because that, in humans and mice, HFD downregulates macrophage EP3 levels and thereby promotes adipogenesis and obesity. Overall, the authors have utilized complementary models to elucidate the important role of the macrophage PGE2-EP3 axis in obesity and demonstrated a well-organized and well-presented study. I have a few comments for further strengthening the current manuscript.

1. I have a conceptual concern that many studies have demonstrated that adipocyte hyperplasia (adipogenesis) is associated with smaller adipocyte size and improved metabolic phenotypes. Authors need to address this apparent inconsistency with the literature.

2. Results showed that deletion of EP3 from myeloid cells (LysM-Cre) results in heavier BAT and reduced energy expenditure. However, the current manuscript majorly focused on WAT and overlooked the contribution by the BAT.
3. In Figure 8, a systemic EP3 agonist was applied to mice. It is unclear whether its anti-obesity effects were majorly contributed by the macrophage EP3-SPARC mechanism.
4. The macrophage function and inflammatory phenotypes were not thoroughly investigated under the condition of myeloid cell-specific deletion of EP3.
5. SPARC is a secreting protein; however, no secreting levels of SPARC have been shown.

Referee #3:

Shang and co-workers report that the prostaglandin E2 receptor subtype 3 (EP3) expression is reduced in adipose tissue macrophages (ATMs) from obese patients and mice fed a high-fat diet (HFD). The authors find that disrupting the EP3 receptor in macrophages exacerbates obesity and promotes adipocyte differentiation by decreasing the expression and secretion of the anti-adipogenic factor SPARC. On the other hand, eliminating SPARC expression undermines the protective effects provided by EP3 signaling. Mechanistically, they demonstrate that EP3 activation enhances SPARC expression by reducing DNA methylation in its promoter via the PKA/Sp1/Dnmt1/3a pathway. Furthermore, pharmacological activation of EP3 with EP3 agonist sulprostone alleviates obesity induced by HFD, emphasizing its potential as a therapeutic target. Overall, the paper is well written and offers interesting findings that are backed by well-executed experiments; however, there are some concerns that need to be addressed to enhance impact and clarity.

Major comments.

1. The disruption of EP3 in macrophages (EP3F/FLysMCre mice) will be a relevant model for the field. It would be beneficial to provide a better phenotypic description of the mice when subjected to high-fat diet (HFD). For example, the authors should consider investigating whether disruption of EP3 in macrophages leads to fibrosis and abnormal angiogenesis in adipose tissue.
2. To strengthen the author's observations that the PGE2/EP3/SPARC axis in macrophages may be involved in adipogenesis, it is important to demonstrate that treatment with PGE2 induces expression of SPARC in wild-type BMDMs, while this effect is not observed in EP3 null BMDMs. Additionally, in the BMDM/3T3-L1 differentiation experiments, the authors should evaluate whether PGE2 inhibits 3T3-L1 differentiation into adipocytes, both with and without the EP3 inhibitor L-798106.
3. What is the inflammatory status of the ATMs under chow diet when EP3 is deleted? Do EP3F/FLysMCre (EP3^{-/-}) mice exhibit increased inflammation in WAT during the HFD diet?
4. The authors report that EP3 deficiency in macrophages worsened high-fat diet (HFD)-induced obesity in mice, leading to enlarged adipocytes and increased expression of fatty acid synthesis and adipogenesis genes (Figure 2). These changes were also observed in EP3F/FLysMCre mice on a normal chow diet (Fig EV1H-M). Does this suggest that a possible obesity phenotype is developing in EP3^{-/-} mice under a chow diet, requiring more time to manifest changes? Should this factor be considered when interpreting results from the study?
5. Do prostaglandin levels in macrophages of epididymal adipose tissue change in EP3^{-/-} mice fed a normal chow diet or HFD?
6. The authors evaluated the expression of EP3 isoforms in palmitate-treated bone marrow-derived macrophages (BMDMs). Analyzing the expression of EP3 isoforms in the macrophages of epididymal adipose tissue under HFD conditions would also be relevant.

Minor comments.

1. Please clarify the duration of BMDM treatment with palmitate in the Figure legends.
2. Including the relative protein densitometric analysis for Figure 1 would be useful.
3. For clarity, mention in the manuscript that a Mac-3 antibody was utilized to identify and visualize adipose tissue macrophages (ATMs).
4. In the immunofluorescence panels displaying the merged fluorescence, please indicate in the corresponding Figure legends what the white arrowheads represent.
5. Figure 1A presents data from human adipose tissue sequence data reported by Emont et al. (2022). The methods section should include a description of how the data was analyzed. In addition, Emont et al. collected expression data from white adipose tissue of mice fed a chow diet and those fed a high-fat diet (HFD). It would be beneficial to examine the expression of PG receptors and SPARC in this context.

Dear Editor,

Thank you very much for your dedication to our manuscript and for giving us the opportunity to revise it. We are also appreciative for the editor's and the reviewers' comments on our manuscript (EMBOJ-2024-119612). We have adopted virtually all these suggestions and made appropriate revisions of the text and figures. We hope it is now suitable for publication in *EMBO J*. Our point-by-point responses (blue) to the reviewers' comments (**black/bold**) are detailed below.

Referee #1:

In this paper, authors tried to characterize the role of macrophage EP3 in adipose tissue. Analyzing macrophage-EP3 depletion model, authors concluded dietary obesity reduced EP3 in macrophage that led to a reduction in SPARC level in this type of cell which aggravated diet-induced obesity. Authors also showed EP3 reduced DNA methylation in macrophage that led to an increase in SPARC level and amelioration of HFD-induced obesity. Major issue relates with the lack of the characterization of inflammation. Recently, SPARC was shown to have pro-inflammatory role in adipose tissue (J Clin Invest. 2023 Oct 2;133(19):e169173. doi: 10.1172/JCI169173., Immunity. 2022. Sep 13;55(9):1609-1626.e7. doi: 10.1016/j.immuni.2022.07.007.). As authors may know, chronic sterile inflammation has critical roles for systemic glucose/ insulin intolerance and unhealthy obesity. These should be characterized in the white adipose tissues of models studied in this paper. Another potential issue is the contribution of macrophage-SPARC in the regulation of adipose SPARC level. The above J Clin Invest paper indicated the level of SPARC as rather low in monocytes compared to adipocytes. It is predicted that the level of SPARC in whole adipose tissue may be regulated mostly by adipocytes, and contribution of macrophage derived SPARC as minor. Authors should show the level of protein SPARC in the visceral fat of the 1) macrophage-EP3 overexpression model, 2) macrophage-EP3 knockout model.

Response: Thank you for the valuable comments.

1) As requested, we examined effect of EP3 deletion on macrophage polarization both *in vitro* and *in vivo* (in fat tissues), and effect of EP3 deletion in macrophages on fat tissue inflammation in mice. As anticipated (*J Cancer Res Clin Oncol*.2023 Aug;149:7053-7067), PA enhanced the M1 polarization and decreased M2 polarization in macrophages (Appendix Fig. S3B-E). However, EP3 deletion did not significantly influence macrophage polarization with and without PA treatment (Appendix Fig. S3B-E). Moreover, EP3 deletion in macrophages did not significantly alter the expression and production of pro-inflammatory cytokines and anti-inflammatory cytokines in the adipose tissues (Appendix Fig. S6A-C) and adipose tissue macrophages (ATMs) from both normal diet and HFD-fed mice (Appendix Fig. S6D-E). We expanded the results, please see the Page 3, Lines 120-123; Page 4, Lines 137-139; and Appendix Figure S3B-E and Appendix Figure S6A-E.

2) Regarding SPARC expression in macrophages, we compared SPARC expression among macrophages, adipocytes and monocytes from human and mice adipose tissues using unbiased single nucleus (sNuc) sequencing data (*Nature*. 2022 Mar;603: 926-933). SPARC expression in macrophages was comparable with that in white adipose tissues from both human and mice (Appendix Figure S8A-B), and SPARC in monocytes is quite low as reported (*J Clin Invest*. 2023 Oct 2;133:e169173). Using RT-PCR assay, we observed that SPARC expression in macrophages was lower than in adipocytes, but barely detected in monocytes (Appendix Figure S8C).

SPARC abundance was markedly reduced in adipose tissues in HFD-fed SPARC^{F/F}LysM^{Cre} mice (Reviewer Figure 1A), further supporting SPARC expression in macrophages. Indeed, EP3 deletion in macrophages reduced, but EP3 overexpression in macrophages increased SPARC expression in adipose tissues in HFD-treated mice (Reviewer Figure 1B-1C). As such, we expanded the macrophage SPARC expression in Results section, please see Page 4, Lines 160-163, and Appendix Figure S8A-C.

Figure for reviewers removed.

Minor issues.

1) In Fig1B, what type of cells or tissues produces PGE₂? What is the mechanism that increases this type of prostaglandin?

Response: Due to very short half-lives, all the PGs, including PGE₂, play their biological effects through binding specific receptors in paracrine and autocrine manner. PGE₂ is synthesized sequentially by cyclooxygenases (COXs) and prostaglandin E synthase (PGES) enzymes. There are three PGES isoforms- microsomal PGES-1 (mPGES-1), mPGES-2, and cytosolic PGES (cPGES). PGE₂ can be generated by most of cells in adipose tissues since PGESs are expressed in most cells (*Nature*. 2022 ;603(7903):926-933.) (Reviewer Figure 2A-B). However, in the inflamed tissues, PGE₂ is dominantly produced in macrophages through inducible COX-2 and mPGES-1(*Int J Biol Sci*. 2023 Aug 6;19(13):4157-4165.). In HFD-induced obese mouse model, COX-2 is upregulated in adipose tissues (*Obesity (Silver Spring)*. 2009 ;17(6):1150-7; *PLoS One*. 2016;11(4):e0153751), along with increased PGE₂ production. We expanded the discussion, please see Pages 7-8, Lines 304-311.

Figure for reviewers removed.

2) Related with Fig. 1C, what is the protein EP3 level in the macrophage of the visceral fat?

Response: This is a good question. We tested different commercial antibodies, there is no good antibodies for this EP3 GPCR protein (*EMBO J.* 2022;41(16):e110439.).

3) In Fig. 1D, how PA reduces EP3 level? The mechanisms need to be shown.

Response: Thanks for the comments. PA upregulates PPAR γ expression in different mammal cells (*Redox Biol.* 2020 Feb;30:101412; *Genes Nutr.* 2018 Jul 6;13:18; *Int J Med Sci.* 2016;13(3):169-78). PPAR γ suppresses EP3 expression via inhibition of κ B activity in macrophages (*PLoS One.* 2014 21;9:e110828). Likewise, we observed PA inhibited EP3 expression in BMDMs by PPAR γ /NF- κ B pathway, silencing PPAR γ attenuated the EP3 downregulation in PA-treated BMDMs. We amended the results, please see Page 3, Lines 102-105, Appendix Figure S1A-D.

4) Fig. 2C and all other panels for GTT that are done under different body weight among genotypes should be done with clamp studies.

Response: Thanks for the comments. As suggested, we performed hyperinsulinemic-euglycemic clamp tests in HFD-fed EP3 mutant mice (both macrophage EP3 deficient and over-expressed mice). We observed similar results as GTT and ITT; please see the revised Figure 2C-D and Appendix Figure S9C-D.

Since it takes long time to obtain sufficient Rosa-EP3 α /SPARC^{F/F}LysM^{Cre} (3 transgenic colonies) and EP^{F/F}Dnmt(1,3a)^{2F/2F}LysM^{Cre} (4 transgenic colonies) mice with at least 3-month HFD treatment for hyperinsulinemic-euglycemic clamp tests, we retained

GTT/ITT results in these mice as recent studies (*EMBO J.* 2024 Nov;43:4846-4869; *EMBO J.* 2024 Aug;43(16):3466-3493; *Cell Metab.* 2025 Mar 26:S1550-4131(25)00105-6; *Cell Metab.* 2025 Mar 4;37(3):656-672.e7).

5) Again, here I would like to emphasize it as critically important to characterize the protein level of SPARC in the visceral fat of the 1) macrophage-EP3 overexpression model, 2) macrophage-EP3 knockout model.

Response: Thanks for the concern on SPARC expression in macrophages. As requested, we have investigated SPARC changes in macrophages from EP3 deficient and overexpression mice (Fig.3E, 3M). Please see the responses to First paragraph and above Reviewer Figure 1A-C.

6) In Fig. 3J, what is the protein SPARC level?

Response: Thanks for the comments. We amended the results, please see **Figure 3M-N** for the updated information.

7) Fig. 5N indicates PPAR-gamma reduced in 3T3-L1 adipocytes administrated with conditioned medium collected from the si-Dnmt treated BMDMs. The following paper indicates inhibition of PPAR-gamma in adipocyte caused insulin resistance (*Proc Natl Acad Sci U S A.* 2003 Dec 23;100(26):15712-7. doi: 10.1073/pnas.2536828100. Epub 2003 Dec 5.), but Fig. 6A, B and D indicate the opposite phenotype. What is the role of adipocyte- PPAR-gamma in this paper? In the core model, PPAR-gamma also needs to be characterized in protein level in cell specific manner.

Response: Thank you for the valuable comment. PPAR γ is a master regulator for the transcriptional activation of adipogenic genes, and controls pre-adipocyte differentiation toward mature adipocytes (*Diabetes.* 2014 Mar;63(3):900-11). In Figure 5, we used pre-adipocyte 3T3-L1 to study the effect of macrophage-derived SPARC on adipocyte differentiation in *in vitro* co-culture system. We observed macrophage EP3 suppressed pre-adipocyte differentiation through SPARC, PPAR γ , Fabp4 and C/EBP α were used as markers of pre-adipocyte differentiation.

The role of PPAR γ in mature adipocytes in insulin sensitivity is still controversial. Evans group reported adipose-specific PPAR γ deficiency (using aP-2^{Cre} mice) causes insulin resistance in fat (*Proc Natl Acad Sci U S A.* 2003 Dec 23;100(26):15712), while Magnuson group reported deletion of PPAR γ in adipose tissues of mice (using aP-2^{Cre} mice) protects HFD-induced insulin resistance (*Proc Natl Acad Sci U S A.* 2005 Apr 15;102(17):6207–6212). Later, Lazar group reported fat-specific PPAR γ knockout mice (adipoq^{Cre} mice) have almost no visible brown and white adipose tissue (*Proc Natl Acad Sci U S A.* 2013 Oct 28;110(46):18656–18661), indicating critical role of PPAR γ in adipogenesis. Thus, PPAR γ in mature adipocytes may have different functions from that in pre-adipocyte progenitor cells.

8) In all figures, figures should be alphabetically demonstrated (for example, not as Fig. 6E, F, H, G, J, K). Group of mice should be clearly labelled in all panels.

Response: We corrected, please see the revised Figure 4H-K, Figure 6G-K, Figure 8H-L.

9) In the Fig. 8M, O, and EV7H, what do these arrows indicate?

Response: Thanks for the comments. In the Fig. 8M, white arrowheads indicated the MAC-3⁺DNMT1⁺ cells. In Fig. 8O, white arrowheads indicated the MAC-3⁺SPARC⁺ cells. In Appendix Figure S12H, white arrowheads indicated the MAC-3⁺DNMT3A⁺ cells. We amended the information; please see Page 24, Lines 1042 and 1045; Appendix Page 15, Lines 327-328.

10) In Fig. EV2B, why the RER increase in the KO mice only at the light time? What is the mechanisms?

Response: We believe, there is the same RER increase trend in EP3 KO mice at both light ($p=0.0188$) and dark time ($P=0.15$). We went through the original data, one extremely readout appeared in KO mice group at dark time (EV2B, Reviewer Figure 3), more animals may be needed for this experiment to reach significance difference.

Figure for reviewers removed.

11) What is the unit for EV3B?

Response:

For single cell RNA sequencing data, the raw gene expression counts (including original EV3B, now Appendix Figure 8E) were subjected to log transformation to stabilize the variance. The log-transformed counts were then used for all subsequent analyses. We mentioned in the method, cited relevant reference (*Nat Methods. 2023 May;20(5):665-672.*); please see Page 14, Lines 576-578.

12) What are the level of proinflammatory cytokines in the studied models? Together with transcriptome studies, some core molecules should be studied in protein levels.

Response: We examined both mRNA and protein levels using RT-PCR and Elisa assay, respectively. EP3 deficiency in macrophages had no significant effects on the expression

of pro-inflammatory cytokines in epididymal adipose tissues and macrophages in HFD-treated mice. We amended the results, please see Page 4, Lines 137-139, Appendix Figure S6A-B and Appendix Figure S6D.

Referee #2:

The manuscript by Shang et al. reported an important role of the PGE2-EP3 axis in macrophages in controlling adipogenesis and obesity pathogenesis. Combining in vitro and various in vivo loss- or gain-of-function mouse models, authors elegantly demonstrated that macrophage-derived PGE2 exerts an autocrine role through EP3 receptor to upregulate SPARC secretion, which exerts downstream anti-adipogenic effects. The underlying molecular mechanisms involved the PKA/SP1/DNMT signaling pathway. This is physiologically important because that, in humans and mice, HFD downregulates macrophage EP3 levels and thereby promotes adipogenesis and obesity. Overall, the authors have utilized complementary models to elucidate the important role of the macrophage PGE2-EP3 axis in obesity and demonstrated a well-organized and well-presented study. I have a few comments for further strengthening the current manuscript.

Response: Thank you very much for the excellent summary of our paper.

1. I have a conceptual concern that many studies have demonstrated that adipocyte hyperplasia (adipogenesis) is associated with smaller adipocyte size and improved metabolic phenotypes. Authors need to address this apparent inconsistency with the literature.

Response: Thank you for the valuable comments.

Obesity is a metabolic state generated by adipose tissue expansion, which occurs through both adipocyte hyperplasia (increase in adipocyte number) and hypertrophy (increase in adipocyte size) (*Cell*. 2022 Feb 3;185(3):419-446). Theoretically, hyperplastic growth is associated with smaller adipocytes accompanied by a better insulin sensitivity, lower inflammation level and less ectopic lipid accumulation (*J Clin Invest*. 2008 Feb;118(2):710-21; *Diabetes*. 2007 Dec;56(12):2910-8; *Diabetologia*. 2000 Dec;43(12):1498-506). Adipocyte hypertrophy leads to dysregulated adipose tissues in obesity characterized by a proinflammatory profile and enhanced insulin resistance (*Nat Rev Mol Cell Biol*. 2019 Apr;20(4):242-258; *Am J Physiol Endocrinol Metab*. 2009 Nov;297(5):E999-E1003). In our study, EP3 deficiency in macrophages promotes adipocytes hyperplasia and hypertrophy in HFD-challenged mice. Likewise, deletion of some metabolic related genes, such as FAS (*Cell Metab*. 2012 Aug 8;16(2):189-201), Abca1 (*Arterioscler Thromb Vasc Biol*. 2018 Apr;38(4):733-743), regulates HFD-induced obesity in mice via interrupting both adipocytes hypertrophy and differentiation. We expanded the discussion, please see Page 6, Lines 242-253.

2. Results showed that deletion of EP3 from myeloid cells (LysM-Cre) results in heavier BAT and reduced energy expenditure. However, the current manuscript majorly focused on WAT and overlooked the contribution by the BAT.

Response: Thanks for the critics. We agree. Indeed, BAT weight and adipocyte size were increased in EP3^{F/F}LysM^{Cre} mice (Figure 2F-G, 2J, 2M). Furthermore, the expression of BAT differentiation markers was decreased, and the expression of WAT differentiation markers were increased in BAT of EP3^{F/F}LysM^{Cre} mice (Figure 2O-P), indicated BAT whitening in HFD-fed EP3^{F/F}LysM^{Cre} mice. We amended the BAT results, please see Page 4, Lines 144-147, Figure 2G, Figure 2J, Figure 2M, Figure 2O-P for more details.

3. In Figure 8, a systemic EP3 agonist was applied to mice. It is unclear whether its anti-obesity effects were majorly contributed by the macrophage EP3-SPARC mechanism.

Response: We agree, and discussed the limitation; please see Page 8, Lines 341-348. Genetic manipulation studies demonstrated EP3 deficiency leads to diet-induced obesity in mice through increase food uptake (*Proc Natl Acad Sci U S A.* 2007 Feb 20;104(8):3009-14), BAT whitening (*EMBO J.* 2022 Aug 16;41(16):e110439) and WAT adipogenesis (*J Mol Cell Biol.* 2016 Dec;8(6):518-529), indicating EP3 involvement of multiple organs/tissues in fat metabolism. In this study, we cannot directly role out the contribution of EP3 in other organs/tissues rather than EP3 in macrophages by administration of EP3 agonist sulprostone. However, induction of SPARC expression, and suppression of its upstream Dnmt1 expression in macrophages were observed in sulprostone-treated animals, indicating macrophage EP3 contribution. Actually, EP3^{F/F}LysM^{Cre} mice were originally planned in the therapeutic settings (Reviewer Figure 4). EP3^{F/F}LysM^{Cre} mice gained heavier than control mice in response to 8-wks HFD feeding (39.51±1.97 vs 34.73±1.95, Reviewer Figure 4). It is really challenge to compare therapeutic effects of sulprostone on HFD-fed mice with different initial body weights.

Figure for reviewers removed.

4. The macrophage function and inflammatory phenotypes were not thoroughly investigated under the condition of myeloid cell-specific deletion of EP3.

Response: As suggested, we examined inflammation phenotypes in BMDMs and mouse tissues. Macrophage-EP3 deficiency had no significant effect on the expression of pro-inflammatory cytokines and anti-inflammatory genes in BMDMs and adipose tissues in mice (Appendix Figure S3B-E, Figure S6A-E). We amended the results, please see Page 3, Lines 120-123; Page 4, Lines 137-139, Appendix Figure S3B-E and Figure S6A-E for more details.

5. SPARC is a secreting protein; however, no secreting levels of SPARC have been shown.

Response: Firstly, SPARC was identified from culture medium supernatant of palmitate-treated WT and EP3^{-/-} BMDMs by secretomic (Figure 3A-D). Secondly, SPARC in culture medium from EP3^{-/-} and WT BMDMs was measured (Figure 3E-F). We amended the results, please see Page 4, Lines 159-160, Figure 3E-F for more details.

Referee #3:

Shang and co-workers report that the prostaglandin E2 receptor subtype 3 (EP3) expression is reduced in adipose tissue macrophages (ATMs) from obese patients and mice fed a high-fat diet (HFD). The authors find that disrupting the EP3 receptor in macrophages exacerbates obesity and promotes adipocyte differentiation by decreasing the expression and secretion of the anti-adipogenic factor SPARC. On the other hand, eliminating SPARC expression undermines the protective effects provided by EP3 signaling. Mechanistically, they demonstrate that EP3 activation enhances SPARC expression by reducing DNA methylation in its promoter via the PKA/Sp1/Dnmt1/3a pathway. Furthermore, pharmacological activation of EP3 with EP3 agonist sulprostone alleviates obesity induced by HFD, emphasizing its potential as a therapeutic target. Overall, the paper is well written and offers interesting findings that are backed by well-executed experiments; however, there are some concerns that need to be addressed to enhance impact and clarity.

Response: We greatly appreciate the reviewer's summarization of the scientific findings revealed in our study.

Major comments.

1. The disruption of EP3 in macrophages (EP3F/FLysMCre mice) will be a relevant model for the field. It would be beneficial to provide a better phenotypic description of the mice when subjected to high-fat diet (HFD). For example, the authors should

consider investigating whether disruption of EP3 in macrophages leads to fibrosis and abnormal angiogenesis in adipose tissue.

Response: As requested, we analyzed fibrosis and abnormal angiogenesis in adipose tissue from HFD-fed EP3^{F/F} and EP3^{F/F}LysM^{Cre} mice. We failed to detect any differences of eWAT fibrosis between EP3^{F/F} mice and EP3^{F/F}LysM^{Cre} mice (Appendix Figure S6F-H). Interestingly, EP3^{F/F}LysM^{Cre} mice exhibited impaired angiogenesis in eWAT compared with EP3^{F/F} mice upon HFD challenge (Appendix Figure S7A-C). We amended the results, please see Page 4, Lines 139-143, Appendix Figure S6F-H and Appendix Figure S7A-C for more details.

2. To strengthen the author's observations that the PGE2/EP3/SPARC axis in macrophages may be involved in adipogenesis, it is important to demonstrate that treatment with PGE2 induces expression of SPARC in wild-type BMDMs, while this effect is not observed in EP3 null BMDMs. Additionally, in the BMDM/3T3-L1 differentiation experiments, the authors should evaluate whether PGE2 inhibits 3T3-L1 differentiation into adipocytes, both with and without the EP3 inhibitor L-798106.

Response: This is a good question. PGE₂ is unstable with very short half-life (several minutes). As requested, we investigated the role of PGE₂ analogue on SPARC expression instead. EP3 agonist Sulprostone, not 17-pt-PGE₂ (EP1 agonist) or Misprostol (EP2/EP4 agonist), upregulated SPARC expression in BMDMs, (Reviewer Figure 5). Moreover, such increase of SPARC induced by salprostone, was attenuated in the EP3^{-/-} BMDMs or by EP3 inhibitor L-798106 (Figure 3H-I). In BMDM/3T3-L1 co-culture system, sulprostone-pretreated BMDMs inhibited 3T3-L1 differentiation toward adipocytes, the inhibitory effect was abrogated in the EP3^{-/-} BMDMs or by EP3 inhibitor L-798106 (Appendix Figure S2A-D). We amended the results, please see Page 3, Lines 111-113; Page 4, Lines 163-165, Figure 3H-I, Appendix Figure S2A-D for more details.

Figure for reviewers removed.

3. What is the inflammatory status of the ATMs under chow diet when EP3 is deleted? Do EP3^{F/F}LysM^{Cre} (EP3^{-/-}) mice exhibit increased inflammation in WAT during the HFD diet?

Response: EP3 deletion in macrophages did not significantly alter the expression and production of pro-inflammatory cytokines and anti-inflammatory cytokines in the adipose tissues (Appendix Fig. S6A-C) and adipose tissue macrophages (ATMs) from both normal

diet and HFD fed-mice (Appendix Fig. S6D-E). We amended the results, please see Page 4, Lines 137-139, Appendix Fig. S6A-E for more details.

4. The authors report that EP3 deficiency in macrophages worsened high-fat diet (HFD)-induced obesity in mice, leading to enlarged adipocytes and increased expression of fatty acid synthesis and adipogenesis genes (Figure 2). These changes were also observed in EP3^{F/F}/LysM^{Cre} mice on a normal chow diet (Fig EV1H-M). Does this suggest that a possible obesity phenotype is developing in EP3^{-/-} mice under a chow diet, requiring more time to manifest changes? Should this factor be considered when interpreting results from the study?

Response: We agree that aging may induce phenotypic changes of EP3^{F/F}LysM^{Cre} mice. Aging eventually triggers chronic low-grade inflammation, adipocyte hypertrophy and insulin resistance in adipose tissues (*Biochem Pharmacol.* 2021 Oct:192:114723). Adult EP3^{F/F}LysM^{Cre} mice had slightly increased body weights on normal chow diet compared to controls (Appendix Fig. S4A), despite normal glucose tolerance, and insulin tolerance. Histologically, enlarged adipocyte size in epididymal and inguinal WAT (Appendix Fig. S4G-K) was observed in EP3^{F/F}LysM^{Cre} mice fed a normal chow diet. As such, we modulated the interpretation, please see Page 4, Lines 135-136.

5. Do prostaglandin levels in macrophages of epididymal adipose tissue change in EP3^{-/-} mice fed a normal chow diet or HFD?

Response: As requested, we tried to measure PG production in ATMs using flow cytometry, but we failed to obtain enough ATMs from epididymal adipose tissues for PG detection. Then we pooled the ATMs and examined PG generation in culture medium (Reviewer Figure 6A). We observed that PGE₂ and PGD₂ were the major PGs in supernatant of BMDMs, and PGE₂, PGD₂, PGF_{2a} and TXB₂ production were increased upon PA treatment. EP3 deletion had no effect on PG generation (Reviewer Figure 6B).

Figure for reviewers removed.

6. The authors evaluated the expression of EP3 isoforms in palmitate-treated bone marrow-derived macrophages (BMDMs). Analyzing the expression of EP3 isoforms in the macrophages of epididymal adipose tissue under HFD conditions would also be relevant.

Response: We did; please see Page 3, Lines 101-102, Figure 1D for the updated information.

Minor comments.

1. Please clarify the duration of BMDM treatment with palmitate in the Figure legends.

Response: We did; please refer to the revised manuscript (Page 21, Line 890 and Page 22, Line 924) for more details.

2. Including the relative protein densitometric analysis for Figure 1 would be useful.

Response: We did; please see Figure 1K-M.

3. For clarity, mention in the manuscript that a Mac-3 antibody was utilized to identify and visualize adipose tissue macrophages (ATMs).

Response: We did, please see Page 24, Lines 1046-1047 for more details.

4. In the immunofluorescence panels displaying the merged fluorescence, please indicate in the corresponding Figure legends what the white arrowheads represent.

Response: Thanks for the comments. In the Fig. 8M, white arrowheads indicated the MAC-3⁺DNMT1⁺ cells. In Fig. 8O, white arrowheads indicated the MAC-3⁺SPARC⁺ cells. In Appendix Figure S12H, white arrowheads indicated the MAC-3⁺DNMT3a⁺ cells. We expanded the figure legend, please see Page 24, Lines 1042 and 1045; Appendix Page 15, Lines 327-328.

5. Figure 1A presents data from human adipose tissue sequence data reported by Emont et al. (2022). The methods section should include a description of how the data was analyzed. In addition, Emont et al. collected expression data from white adipose tissue of mice fed a chow diet and those fed a high-fat diet (HFD). It would be beneficial to examine the expression of PG receptors and SPARC in this context.

Response: We apologize for the negligence. We add data analysis to Method, please refer to the revised manuscript (Page 14, Lines 572-581) for more details.

We analyzed the alterations in PG receptors and *SPARC* expression in macrophages from adipose tissues of HFD-fed mice. Both EP3 and *SPARC* expression levels trend to be downregulated in macrophages of inguinal white adipose tissues after HFD feeding (13

weeks) compared with NCD feeding (Reviewer Figure 7A), but for unknown technical reasons, insufficient macrophages were detected for statistical analysis. Therefore, we further analyzed the alterations in macrophage *EP3* and *SPARC* expression in white adipose tissues after HFD feeding at different time points (*Cell. 2019 Jul 25;178(3):686-698.e14*). Similar to the alterations observed in human subjects, *SPARC* were gradually downregulated in macrophages of epididymal visceral adipose tissue in mice after 6, 12, and 18 weeks of HFD feeding, and was significantly decreased at 18 weeks of HFD feeding (Reviewer Figure 7B), while expression of *EP3* in macrophages was not detected.

Figure for reviewers removed.

Dear Dr Yu,

Thank you for submitting your revised manuscript (EMBOJ-2024-119612R) to The EMBO Journal, as well for your patience with our response. Your amended study was sent back to the three referees for their scientific reassessment, and we have received re-reports from all of them, which I enclose below. As you will see, the experts state that the work has been substantially enhanced by the revisions and they are now broadly in favour of publication.

Thus, we are pleased to inform you that your manuscript has been accepted in principle for publication in The EMBO Journal.

We now need you to take care of a number of minor issues related to formatting and data annotation as detailed below, which need to be addressed at resubmission of the work.

Please contact me at any time if you have additional questions related.

As you might have noted from our webpage, every paper at the EMBO Journal now includes a 'Synopsis', displayed on the html and freely accessible to all readers. The synopsis includes a 'model' figure as well as 2-5 one-short-sentence bullet points that summarize the article. I would appreciate if you could provide this figure and related text highlights.

Thank you for giving us the chance to consider your manuscript for The EMBO Journal. I look forward to your final revision.

Again, please contact me at any time if you need any help or have further questions.

Kind regards,

Daniel Klimmeck

>> Author Contributions: Remove the author contributions information from the manuscript text. Note that CRediT has replaced the traditional author contributions section as of now because it offers a systematic machine-readable author contributions format that allows for more effective research assessment. and use the free text boxes beneath each contributing author's name to add specific details on the author's contribution.

More information is available in our guide to authors.
<https://www.embopress.org/page/journal/14602075/authorguide>

>>Appendix file with ToC: please remove the red font and line numbers.

>> Data availability section: please remove the referee access information and token and make sure privacy is released from the PRIDE dataset.

>> Section order should be corrected as follows: title page with complete author information, abstract, keywords, introduction, results, discussion, methods, data availability section, acknowledgements, disclosure and competing interests statement, references, main figure legends, tables, expanded figure legends.

>> Consider additional changes and comments from our production team as indicated below:
- Figure legends:

1. Please note that the exact p values are not provided in the legends of figures 1A-E, H, I, K-M; 2A, C, F, H-P; 3F, G, J, K, L, N, P, Q, R, S; 4A, C, E, F, H, I, L, M, N, O, P; 5C, D, F, J, K, L, M; 6A, C, E, F, H, I, L, M, N, O, P; 7B, C, E, F; 8B, D, F, G, I, J, K, L, N, P.

2. Please note that the scale bar needs to be defined for figures 2G, 4G, 5G

- typos: please correct l.368 'Regents' to 'Reagents'.

Referee #1:

Authors mostly responded to the issues raised. I have no further questions or comments.

Referee #2:

The revision adequately addresses my previous concerns.

Referee #3:

The authors have adequately addressed all comments arising from the original submission, incorporating additional experiments that clarify the role of PGE2 and EP3 in the protective effects of the PGE2/EP3/SPARC axis on macrophages in the context of obesity. Overall, the manuscript shows significant improvements in both experimental consistency and discussion. I recommend that this manuscript be published.

The authors addressed the remaining editorial issues.

Dear Dr Yu,

Thank you for submitting the revised version of your manuscript. I have now evaluated your amended manuscript and concluded that the remaining minor concerns have been sufficiently addressed.

I am thus pleased to inform you that your manuscript has been accepted for publication in the EMBO Journal.

On a different note, I would like to alert you that EMBO Press offers a format for a video-synopsis of work published with us, which essentially is a short, author-generated film explaining the core findings in hand drawings, and, as we believe, can be very useful to increase visibility of the work. Please see the following link for representative examples and their integration into the article web page:

<https://www.embopress.org/doi/full/10.15252/emj.2019103932>

Best regards,

Daniel Klimmeck

Daniel Klimmeck, PhD
Senior Editor
The EMBO Journal
EMBO
Postfach 1022-40
Meyerhofstrasse 1
D-69117 Heidelberg
contact@embojournal.org